

# Effects of Arctic stratospheric ozone changes on spring precipitation in the northwestern United States

Xuan Ma[1], Fei Xie[1*], Jianping Li[1,2], Wenshou Tian[3], Ruiqiang Ding[4],

Cheng Sun[1], and Jiankai Zhang[3]

[1]*State Key Laboratory of Earth Surface Processes and Resource Ecology and College of Global*

*Change and Earth System Science, Beijing Normal University, Beijing, China*

[2]*Laboratory for Regional Oceanography and Numerical Modeling, Qingdao National Laboratory*

*for Marine Science and Technology, Qingdao, China*

[3]*College of Atmospheric Sciences, Lanzhou University, Lanzhou, China*

[4]*State Key Laboratory of Numerical Modeling for Atmospheric Sciences and Geophysical Fluid*

*Dynamics, Institute of Atmospheric Physics, Chinese Academy of Sciences, Beijing, China*

Submitted as an Article to: ***Atmospheric Chemistry and Physics***

*7 June 2018*

* Corresponding author:

Dr. Fei Xie, Email: xiefei@bnu.edu.cn.



# 1 Abstract

Using observations and reanalysis, we find that changes in April precipitation
variations in the northwestern US are strongly linked to March Arctic stratospheric
ozone (ASO). An increase (decrease) in ASO can result in enhanced (weakened)
westerlies in the high and low latitudes of the North Pacific but weakened (enhanced)
westerlies in the mid-latitudes. The anomalous circulation over the North Pacific can
extend eastward to western North America, facilitating (impeding) the flow of a dry
and cold airstream from the middle of North America to the North Pacific and
enhancing (weakening) downwelling in the northwestern US, which results in
decreased (increased) precipitation there. Model simulations using WACCM4 support
the statistical analysis of observations and reanalysis data, and further reveal that the
ASO influences circulation anomalies over the northwestern US in two ways.
Stratospheric circulation anomalies caused by the ASO changes can propagate
downward to the troposphere in the North Pacific and then eastward to influence the
strength of the circulation anomalies over the northwestern US. In addition, the ASO
changes cause sea surface temperature anomalies over the North Pacific that would
cooperate with the ASO changes to modify the circulation anomalies over the
northwestern US. Our results suggest that ASO variations could be a useful predictor
of spring precipitation changes in the northwestern US; The northwestern US may
become dryer in future springs due to ASO recovery.



## 1. Introduction

Stratospheric circulation anomalies can affect tropospheric climate via chemical–radiative–dynamical feedback processes (Baldwin and Dunkerton, 2001; Graf and Walter, 2005; Cagnazzo and Manzini, 2009; Ineson and Scaife, 2009; Thompson et al., 2011; Reichler et al., 2012; Karpechko et al., 2014; Kidston et al., 2015; Li et al., 2016; Zhang et al., 2016; Wang et al., 2017). Since stratospheric ozone can influence stratospheric circulation via the atmospheric radiation balance (Tung, 1986; Haigh, 1994; Ramaswamy et al., 1996; Forster and Shine, 1997; Pawson and Naujokat, 1999; Randel and Wu, 1999, 2007; Solomon, 1999; Labitzke and Naujokat, 2000), the impact of polar ozone on tropospheric climate change has recently received widespread attention.

In recent decades, Antarctic stratospheric ozone has decreased dramatically due to the increase in anthropogenic emissions of ozone depleting substances (Solomon, 1990, 1999; Ravishankara et al., 1994, 2009). Numerous studies have found that the decreased Antarctic ozone has contributed substantially to climate change in the Southern Hemisphere. The Southern Hemisphere circulation underwent a marked change during the early 21st century, with a slight poleward shift of the westerly jet (Thompson and Solomon, 2002; Lemke et al., 2007). Subsequent studies concluded that Antarctic ozone depletion is responsible for at least 50% of the circulation shift (Lu et al., 2009; Son et al., 2010; McLandress et al., 2011; Polvani et al., 2011; Hu et al., 2013; Gerber and Son, 2014; Waugh et al., 2015). In addition, the poleward displacement of the westerly jet has been linked to an extension of the Hadley cell (Son et al., 2009, 2010; Min and Son, 2013) and variations in mid- to high-latitude precipitation during austral summer; i.e., increased rainfall in the mid-latitudes and reduced rainfall in the high latitudes of the Southern Hemisphere (Thompson et al.,



2000, 2011; Marshall, 2003; Archer and Caldeira, 2008; Fogt et al., 2009; Son et al.,
2009; Feldstein, 2011; Kang et al., 2011; Polvani et al., 2011). The changes in
Antarctic ozone are not only related to the displacement of the westerly jet in the
Southern Hemisphere, but also affect its intensity. Thompson and Solomon (2002)
argued that Antarctic ozone depletion can also enhance westerly winds via the strong
radiative cooling effect and thermal wind relationship. The westerly winds are
enhanced from the stratosphere to the mid-latitude troposphere in the case of wave–
mean flow interaction (Son et al., 2010; Thompson et al., 2011), thereby accelerating
circumpolar currents in the mid-latitudes. The changes in near-surface circumpolar
currents restrict the spread of polar cold air to lower latitudes, causing evident climate
cooling in the Antarctic interior and warming in the mid-latitudes and subpolar
regions. Moreover, changes in subtropical drought, storm tracks and ocean circulation
in the Southern Hemisphere are closely related to Antarctic ozone variations (Yin,
2005; Russell et al., 2006; Son et al., 2009; Polvani et al., 2011; Bitz and Polvani,

60  2012).

The variations in Arctic stratospheric ozone (ASO) in the past five decades are
quite different from those of Antarctic stratospheric ozone, as the multi-decadal loss
of ASO is much smaller than that of Antarctic stratospheric ozone (WMO, 2011).
However, sudden stratospheric warming in the Arctic (Randel, 1988; Charlton and
Polvani, 2007; Manney et al., 2011; Manney and Lawrence, 2016) means that the
year-to-year variability in ASO has an amplitude equal to or even larger than that of
Antarctic stratospheric ozone. Thus, the effect of ASO on Northern Hemisphere
climate change has also become a matter of concern.
The depletion of ASO can cause circulation anomalies, corresponding to the



positive polarity of the Northern Annular Mode (NAM)/North Atlantic Oscillation
(NAO) that can affect tropospheric climate and the incidence of extreme weather
events. Cheung et al. (2014) used the UK Met Office operational weather forecasting
system and Karpechko et al. (2014) used ECHAM5 simulations to investigate the
relationship between extreme Arctic ozone anomalies in 2011 and tropospheric
climate. Smith and Polvani (2014) used an atmospheric global climate model to reveal
a significant influence of ASO changes on tropospheric circulation, surface
temperature, and precipitation when the amplitudes of the forcing ASO anomaly in
the model are larger than those historically observed. Subsequently, using a fully
coupled chemistry–climate model, Calvo et al. (2015) again confirmed that changes in
ASO can produce robust anomalies in Northern Hemisphere temperature, wind, and
precipitation. Furthermore, the effects of ASO on the Northern Hemisphere climate
can be seen in observations. Ivy et al. (2017) presented observational evidence for the
relationship between ASO and tropospheric climate, revealing that the maximum
daily surface temperature anomalies in spring (March–April) in some regions of the
Northern Hemisphere occurred during years with low ASO in March. Xie et al. (2016,
2017a, 2017b) demonstrated that the tropical climate can also be affected by ASO.
They pointed out that stratospheric circulation anomalies caused by March ASO
changes can rapidly extend to the lower troposphere and then propagate horizontally
to the North Pacific in about 1 month, influencing the North Pacific sea surface
temperature (SST) in April. The induced SST anomalies (Victoria Mode) associated
with the circulation anomalies can influence El Niño–Southern Oscillation (ENSO)
and tropical rainfall over a timescale of ~20 months.
As shown above, a large number of observations and simulations have shown
that ASO variations have a significant impact on Northern Hemisphere tropospheric



climate, but few studies have focused on regional characteristics. Xie et al. (2018)
found that the ASO variations could significantly influence rainfall in the central of
China, since the circulation anomalies over the North Pacific caused by ASO
variations can extend westward to China. This motivates us to investigate whether the
circulation anomalies extend eastward to affect the precipitation in North America. In
this study, we find a strong link between ASO and precipitation in the northwestern
US in spring. We focus on analyzing the characteristics of the impact of ASO on
precipitation in the northwestern US in spring and the associated mechanisms. The
remainder of this manuscript is organized as follows. Section 2 describes the data and
numerical simulations, and section 3 discusses the relationship between the ASO
anomalies and precipitation variations in the northwestern US, as well as the
underlying mechanisms. The results of simulations are presented in section 4, and
conclusions are given in section 5.
**2.   Data and simulations**
The ASO variations is defined as the Arctic stratospheric ozone averaged over the
latitude of 60°–90°N at an altitude of 100–50 hPa after removing the seasonal cycle
and trend. Ozone values used in the present analysis are derived from the
Stratospheric Water and OzOne Satellite Homogenized (SWOOSH) dataset (Davis et
al., 2016), which is a collection of stratospheric ozone and water vapor measurements
obtained by multiple limb sounding and solar occultation satellites over the previous
30 years. Monthly mean ozone data from SWOOSH (1984–2016) is zonal–mean
gridded dataset at a horizontal resolution of 2.5° (latitude: 89°S to 89°N) and vertical
pressure range of 31 levels from 316 hPa to 1 hPa. Another set of ozone dataset is
taken from Global Ozone Chemistry and Related trace gas Data Records for the
Stratosphere (GOZCARDS, 1984–2013) project (Froidevaux et al., 2015) based on



high quality data from past missions (e.g., SAGE, HALOE data) and ongoing
missions (ACE-FTS and Aura MLS). It is also a zonal–mean dataset with a
meridional resolution of 10°, extending from the surface to 0.1 hPa (25 levels).
In addition, two sets of global precipitation reanalysis datasets are employed in
this study: monthly mean precipitation data constructed by the Global Precipitation
Climatology Project (GPCP), which is established by the World Climate Research
program (WCRP) in 1986 aiming to observe and estimate the spatial and temporal
global precipitation (Huffman et al., 1997), with a resolution of 2.5° latitude/longitude
grid for the analysis period 1984–2016; global terrestrial rainfall dataset derived from
the Global Precipitation Climatology Centre (GPCC) based on quality-controlled data
from 67200 stations world-wide, with a resolution of 1.0° latitude/longitude grid. In
addition, SST is taken from the UK Met Office Hadley Centre for Climate Prediction
and Research SST (HadSST). Other atmospheric datasets including monthly-mean
wind and geopotential height fields for the period 1984–2016 are obtained from the
NCEP/Department of Energy (DOE) Reanalysis 2 (NCEP-2), regarding as an updated
NCEP/NCAR Reanalysis Project (NCEP-1).
We use the Whole Atmosphere Community Climate Model version 4
(WACCM4), a part of the National Center for Atmospheric Research's Community
Earth System Model (CESM), version 1.0.6, to investigate precipitation response in
the northwestern U.S. to the ASO anomalies. WACCM4 encompasses the Community
Atmospheric Model version 4 (CAM4) and as such includes all of its physical
parameterizations (Neale et al., 2013). It uses a coupled system made up of four
components, namely atmosphere, ocean (specified SST), land, and sea ice (Holland et
al., 2012) and has detailed middle–atmosphere chemistry. This improved version of



WACCM uses a finite-volume dynamical core, and it extends from the surface to
approximately 145 km geometric altitude (66 levels), with a vertical resolution of
about 1 km in the tropical tropopause layer and the lower stratosphere. The
simulations in the present paper is at a $1.9° \times 2.5°$ horizontal resolution, and do not
include interactive chemistry (Garcia et al., 2007). More information can be seen in
Marsh et al. (2013). The model's radiation scheme uses these conditions: fixed
greenhouse gas (GHG) values, averages of emissions scenario A2 of the
Intergovernmental Panel on Climate Change (IPCC) (WMO, 2003) for 1980–2015.
The prescribed ozone forcing used in the experiments is a 12–month seasonal cycle
averaged over the period 1980–2015 from CMIP5 ensemble mean ozone output. The
Quasi Biennial Oscillation (QBO) phase signals with a 28–month fixed cycle are
included in WACCM4 as an external forcing for zonal wind.
Seven time–slice experiments (R1–R7) are designed to investigate the
precipitation changes in the northwestern U.S. due to the ASO anomalies. Details of
the seven experiments are given in Table 1. All the experiments are run for 33 years,
with the first 3 years excluded for the model spin–up and only the last 30 years are
used for analysis.
**3.   Response of precipitation in the northwestern US to ASO anomalies in**
**spring**
Since the variations in ASO are most obvious in March due to the Arctic polar vortex
rupture (Manney et al., 2011), previous studies have reported that the ASO changes in
March have the strongest influence on the Northern Hemisphere (Ivy et al., 2017; Xie
et al., 2017a). In addition, these studies pointed out that the changes in ASO affect the
tropospheric climate with a lead of about 1–2 months; the relevant mechanisms have



been investigated in detail by Xie et al. (2017a). We therefore show in Fig. 1 the
correlation coefficients between ASO variations in March from SWOOSH and
GOZCARDS data, and precipitation anomalies in April from GPCC and GPCP data
over western North America. In all cases in Fig. 1 the March ASO changes are
significantly anti-correlated with April precipitation anomalies in the northwestern US
(mainly in Washington and Oregon states), implying that positive (negative) spring
ASO anomalies are associated with less (more) spring precipitation in the
northwestern US. Note that since this kind of feature appears in the northwestern U.S.,
the Fig. 1 shows only the west side of North America.

The correlation coefficients between March ASO variations and precipitation

anomalies (January to December are in the same year) in the northwestern US are
shown in Fig. 2. The correlation coefficients between March ASO variations and April
precipitation anomalies in the northwestern US are the largest and are significant at
the 95% confidence level. Note that the correlation coefficients between March ASO
variations and July precipitation anomalies are also significant. The impact of March
ASO on the northwestern US in summer and the associated mechanisms are different
from those considered in this study (not shown) and will be presented in another paper,
but will not be investigated further here. March ASO changes are not significantly
correlated with simultaneous (March) precipitation variations (Fig. 2), illustrating that
the ASO changes lead precipitation anomalies by about 1 month. Since the results
from four sets of observations show a common feature, and SWOOSH and GPCP data
span a longer period, only SWOOSH ozone and GPCP precipitation are used in the
following analysis.

Figure 3 shows the differences between composite anomalies of April



precipitation in the northwestern US during positive and negative March ASO
anomaly events to further confirm the relationship between March ASO changes and
April precipitation anomalies in the northwestern US. The increase (decrease) in
March ASO is associated with decreased (increased) April precipitation in the
northwestern US. The pattern of the difference in Fig. 3 is consistent with the pattern
of the correlation coefficients in Fig. 1. Note that an anomalous signal of precipitation
is found in the southwestern US; however, this phenomenon is not significant in Fig.
1 and may be an artefact of the composite analysis.

The above statistical analysis shows a strong negative correlation between March

ASO variations and April precipitation anomalies in the northwestern US, meaning
that the ASO can be used to predict changes in spring precipitation in the
northwestern US. It may also imply that the northwestern US will become dryer in
future springs due to ASO recovery. The process and underlying mechanism that are
responsible for the impact of ASO anomalies on precipitation changes need further
analysis.

Figure 4a, c and e shows the correlation coefficients between March ASO

anomalies and April zonal wind variations at 200, 500, and 850 hPa, respectively. The
spatial distribution of significant correlation coefficients over the North Pacific
exhibits a tripolar mode with a zonal distribution at 200 and 500 hPa; i.e. a positive
correlation in the high and low latitudes in the North Pacific and a negative
correlation in mid-latitudes. This implies that the increase (decrease) in ASO can
result in enhanced (weakened) westerlies in the high and low latitudes of the North
Pacific but weakened (enhanced) westerlies in the mid-latitudes. At 850 hPa, the
anomalous circulation signal in the low latitudes of the North Pacific has weakened



and disappeared. It is evident from Fig. 4 that the anomalous changes in the zonal
wind over the North Pacific can extend westward to East Asia. Xie et al. (2018)
identified the effect of spring ASO changes on spring precipitation in China. Note that
the zonal wind anomalies can also extend eastward to North America, implying a
possible influence of spring ASO variations on weather and climate in western North
America. Figure 4b, d and f shows the differences between April zonal wind
anomalies during positive and negative March ASO anomaly events at 200, 500, and
850 hPa, respectively. The spatial distributions of the differences are in good
agreement with those of the correlation coefficients between March ASO and April
zonal wind variations (Fig. 4a, c and e). Figure 5a, c and e shows the correlation
coefficients between March ASO and April geopotential height variations at 200, 500,
and 850 hPa, respectively. The differences between April geopotential height
anomalies during positive and negative March ASO anomaly events at 200, 500, and
850 hPa are shown in Fig. 5b, d and f, respectively. The variations in geopotential
height associated with ASO anomalies correspond closely to those of zonal wind.

Figures 4 and 5 show that the March ASO anomalies may be linked with

anomalous April zonal wind over the North Pacific; i.e., the increase (decrease) in
ASO can result in enhanced (weakened) westerlies in the high and low latitudes of the
North Pacific but weakened (enhanced) westerlies in the mid-latitudes. It is clear that
the weakened (enhanced) westerlies in the mid-latitudes and the enhanced (weakened)
westerlies in the low latitudes can extend eastward to the western US. This kind of
circulation anomaly corresponds to an anomalous cyclone (anticyclone) in the western
US in the middle and upper troposphere, which is likely associated with a strong low
(high) pressure system in the middle and upper troposphere and a relatively weak high
(low) pressure system in the lower troposphere.



To further validate our conjecture regarding the response of the circulation in the
western US to ASO changes, we analyze the differences between April horizontal
wind anomalies during positive and negative March ASO anomaly events at 200, 500,
and 850 hPa (Fig. 6). There is an anomalous cyclone in the southwestern US related
to the increase in March ASO. This kind of circulation anomaly over the southwestern
US enhances cold and dry airflow from the North American continent to the North
Pacific, reducing the water vapor concentration in the air over the northwestern US,
and possibly decreasing the local April precipitation. In addition, a strong
low-pressure system in the middle and upper troposphere over the western US during
positive ASO anomaly events (Fig. 6) suggests downwelling flow in the region.
Figure 7 shows a longitude–latitude cross-section of differences between April
vertical velocity anomalies averaged over 1000–500 hPa during positive and negative
March ASO anomaly events. When the March ASO increases, tropospheric
convective activity in the northwestern US (115°–130° W) weakens, corresponding to
anomalous downwelling. This situation may also decrease April precipitation in the
northwestern US. Figure 7b depicts a longitude–height cross-section of differences
between April vertical velocity averaged over 43°–50°N during positive and negative
March ASO anomaly events, which further shows that tropospheric convective
activity over the northwestern US enhances when the March ASO increases. Based on
the above analysis, the circulation anomalies in the northwestern US associated with
positive (negative) March ASO anomalies may reduce (increase) the local
precipitation in April.
**4.   Simulations of the effect of ASO variations on precipitation in the**
**northwestern US during spring**





Using observations and reanalysis data, we investigated the relationship between
March ASO and April precipitation in the northwestern US and revealed the
underlying mechanisms in section 3. In this section, we use WACCM4 simulations
(see section 2) to confirm the above conclusions. First, we check the model
performance in simulating precipitation over western North America. Figure 8 shows
the April precipitation climatology over the region 95°–140°W, 30°–63°N from the
control experiment R1 (Table 1) and from GPCP for the period 1995–2005. The
model simulates a center of high precipitation over the west coast of North America
(Fig. 8a). It is clear that the spatial distribution of the simulated precipitation
climatology is similar to that calculated by GPCP (Fig. 8b).
Figure 9a displays the differences in April precipitation between experiments R3
and R2. The pattern of simulated April precipitation anomalies forced by ASO
changes in western North America (Fig. 9a) is nearly opposite to that observed (Fig.
3); i.e., the increased March ASO forces an increase in precipitation in the
northwestern US. The differences in April zonal wind at 200, 500, and 850 hPa
between experiments R3 and R2 are shown in Fig. 9b, c, and d, respectively. The
simulated pattern of April zonal wind anomalies in western North America (Fig. 9b, c
and d) is also opposite to that observed (Fig. 4). Comparing the global pattern of
simulated April zonal wind anomalies with the observations, it is surprising to find
that the positions of simulated zonal wind anomalies over the Northeast Pacific and
western North America are shifted northward. This results in the simulated
precipitation anomalies over western North America also shifting northward, so that a
decrease in precipitation on the west coast of Canada in April is found in Fig. 9a. This
explains why we find the pattern of simulated April precipitation anomalies in the
North America (Fig. 9a) is nearly opposite to that observed (Fig. 3). Figure 9 shows





that the results of the model simulation in which we only change the ASO forcing do
not reflect the real situation of April precipitation anomalies in the northwestern US,
with a shift in position compared with observations. This leads us to consider whether
other factors interact with March ozone to influence April precipitation in the
northwestern US.
Previous studies have found that the North Pacific SST has a significant effect on
precipitation in the US (e.g., Namias, 1983; Ting and Wang, 1997; Wang and Ting,
2000; Barlow et al., 2001; Lau et al., 2002; Wang et al., 2014). Figure 10a shows the
correlation coefficients between regional averaged (43°–50°N, 115°–130°W)
precipitation anomalies and SST variations in April. Interestingly, the results show
that the distribution of correlation coefficients over the North Pacific has a meridional
tripole structure, which is referred to as the Victoria Mode SST anomaly pattern. Xie
et al. (2017a) reported that the stratospheric circulation anomalies caused by March
ASO changes can rapidly extend to the North Pacific over about 1 month, influencing
the North Pacific SST and inducing Victoria Mode anomalies. Figure 10b shows the
correlation coefficients between March ASO (multiplied by –1) and April SST
variations. The pattern in Fig. 10b is in good agreement with that in Fig. 10a. It is
further found that removing the Victoria Mode signal from the time series of
precipitation in the northwestern US reduces the correlation coefficient between
March ASO anomalies and filtered April precipitation variations in the northwestern
US to –0.40 (the correlation coefficient is –0.63 for the original time series, see Fig.
2), but it remains significant. Figure 10 indicates that the ASO possibly influences
precipitation anomalies in the northwestern US in two ways. First, the stratospheric
circulation anomalies caused by the ASO changes can propagate downward to the
North Pacific troposphere and eastward to influence precipitation over northwestern





US. Second, the ASO changes generate SST anomalies over the North Pacific that act
as a bridge for ASO to affect precipitation in the northwestern US. The SST
anomalies caused by ASO change likely interact with the direct changes in
atmospheric circulation driven by the ASO change to jointly influence precipitation in
the northwestern US. Experiments R2 and R3 do not include the effects of SST, which
may explain why the results of the model simulation in which we only change the
ASO forcing do not reflect the observed precipitation anomalies in the northwestern
US (Fig. 9).

Two sets of experiments (R4 and R5) that include the joint effects of ASO and

SST change are added. Details of the experiments are given in Table 1. Figure 11
shows the differences in April precipitation and zonal wind between experiments R5
and R4. It is clear that the simulated changes in precipitation in the northwestern US
(Fig. 11a) are in good agreement with the observed anomalies shown in Fig. 3; i.e.,
the increase in March ASO forces a decrease in April precipitation in the northwestern
US. In addition, the spatial distributions of simulated zonal wind anomalies (Fig. 11b–
d) are consistent with the observations (Fig. 4). Overall, the simulated precipitation
and circulation in R4 and R5 are no longer shifted northward and are closer to the
observations.

To further emphasize the importance of the joint effects of ASO and ASO-related

SST anomalies on precipitation in the northwestern US, we investigate whether the
spring Victoria Mode-like SST anomalies alone could force the observed
precipitation anomalies in the northwestern US. Two sets of experiments are
performed here (R6 and R7), in which only April SST anomalies over the North
Pacific have been changed (Fig. 12). Details of the experiments are given in Table 1.



Figure 13 shows the differences in April precipitation and zonal wind between
experiments R7 and R6. The simulated precipitation anomalies over the west coast of
the US (Fig. 13a) are much weaker than in the observations (Fig. 3), and the
simulated circulation anomalies (Fig. 13b–d) are quite different from those in Fig. 4.
This suggests that the ASO-related North Pacific SST anomalies alone cannot force
the observed precipitation anomalies in the northwestern US, but that the combined
effect of ASO and ASO-related North Pacific SST anomalies is required (Fig. 11).
Thus, we have shown that the relationship between March ASO and April
precipitation in the northwestern US in the observations and the underlying
mechanisms can be verified by WACCM4.
**5. Summary and conclusions**
Many observations and simulations have shown that ASO variations have a
significant impact on Northern Hemisphere tropospheric climate, but few studies have
focused on regional characteristics. Using observations, reanalysis datasets, and
WACCM4, we have shown that spring ASO changes have a significant effect on April
precipitation in the northwestern US (mainly in Washington and Oregon states) with a
lead of 1–2 months. When the March ASO is anomalously high (low), April
precipitation decreases (increases) in the northwestern US.
During positive ASO events, the zonal wind changes over the North Pacific
exhibit a tripolar mode with a zonal distribution accompanied by geopotential height
anomalies; i.e., enhanced westerlies in the high and low latitudes of the North Pacific,
and weakened westerlies in the mid-latitudes. The anomalous wind can extend
eastward to North America, causing anomalous circulation in western North America.
Such circulation anomalies force an anomalous cyclone in the western US in the





middle and upper troposphere, which likely enhances cold and dry airflow from the
North American continent to the North Pacific, reducing the water vapor
concentration in the air over the northwestern US. At the same time, convection in the
northwestern US is weakened. The two processes possibly decrease April
precipitation in the northwestern US. When the March ASO is reduced, the effect is
just the opposite.
The WACCM4 model is used to confirm the statistical results of observations
and the reanalysis data. The results of the model simulation in which we only change
the ASO forcing do not reflect the observed precipitation anomalies in the
northwestern US in April; i.e., the pattern of simulated April precipitation and
circulation anomalies in the western North America is opposite to that observed. It is
found that SST anomalies over North Pacific caused by ASO changes are likely to
interact with ASO changes to jointly influence precipitation in the northwestern US.
Thus, the ASO influences precipitation anomalies over the northwestern US in two
ways. First, the stratospheric circulation anomalies caused by the ASO change can
propagate downward to the North Pacific troposphere and directly influence
precipitation over the northwestern US. Second, the ASO changes generate SST
anomalies over the North Pacific that act as a bridge, allowing the ASO changes to
affect precipitation in the northwestern US.
**Acknowledgments.** Funding for this project was provided by the National Natural
Science Foundation of China (41575039 and 41630421). We acknowledge ozone
datasets from the SWOOSH and GOZCARDS; precipitation from China
Meteorological Administration, GPCC and GPCP; Meteorological fields from NCEP2,
and WACCM4 from NCAR.



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



**Table 1**. CESM-WACCM4 experiments with various specified ozone and SST
forcing.

| Exp[*1] | Specified ozone and SST forcing | Other forcing |
|---|---|---|
| R1 | Time-slice run as the control experiment used case F_2000_WACCM_SC. The specified ozone forcing is a 12-month cycle of monthly ozone averaged from 1995 to 2005. The specified SST forcing is a 12-month cycle of monthly SST averaged from 1995 to 2005. | Fixed solar constant, fixed greenhouse gas (GHG) values (averages of emissions scenario A2 of the Intergovernmental Panel on Climate Change (WMO, 2003) over the period 1995–2005), volcanic aerosols (from the Stratospheric Processes and their Role in Climate (SPARC) Chemistry–Climate Model Validation (CCMVal) REF-B2 scenario recommendations), and QBO phase signals with a 28-month zonal wind fixed cycle. |
| R2 | Same as R1, except that the March ozone in the region 30°–90°N at 300–30 hPa[*2] is decreased by 15% compared with R1. | Same as R1 |
| R3 | Same as R1, except that March ozone in the region 30°–90°N at 300–30 hPa is increased by 15% compared with R1. | Same as R1 |
| R4 | Same as R2, except that a SST anomalies in the region 0°–70°N and 120°E–90°W related to negative ASO anomalies[*3] is added in the SST forcing in April. | Same as R1 |
| R5 | Same as R3, except that a SST anomalies in the region 0°–70°N and 120°E–90°W related to positive ASO anomalies[*4] is added in the SST forcing in April. | Same as R1 |
| R6 | Same as R1, except that a SST anomalies in the region 0°–70°N and 120°E–90°W related to negative ASO anomalies[*3] is added in the SST forcing in April. | Same as R1 |





| | | | |
|---|---|---|---|
| R7 | Same as R1, except that a SST anomalies in the region 0°–70°N and 120°E–90°W related to positive ASO anomalies[*4] is added in the SST forcing in April. | Same as R1 | |

[*1]Integration time for time-slice runs is 33 years.
[*2]To avoid the effect of the boundary of ozone change on the Arctic stratospheric
circulation simulation, the replaced region (30°–90°N, 300–30 hPa) was larger than
the region used to define the ASO index (60°–90°N, 150–50 hPa).
[*3]For SST anomalies, see Fig. 12a.
[*4]For SST anomalies, see Fig. 12b.



**Table 2.** Selected positive and negative years for March ASO anomaly events based
on SWOOSH data for the period 1984–2016. Positive and negative March ASO
anomaly events are defined using a normalized time series of March ASO variations
from 1984 to 2016. Values larger than 1 standard deviation are defined as positive
March ASO anomaly events, and those below –1 standard deviation are defined as
negative March ASO anomaly events.

| Positive March ASO anomaly events | Negative March ASO anomaly events |
| --- | --- |
| 1998, 1999, 2001, 2004, 2010 | 1993, 1995, 1996, 2000, 2011 |



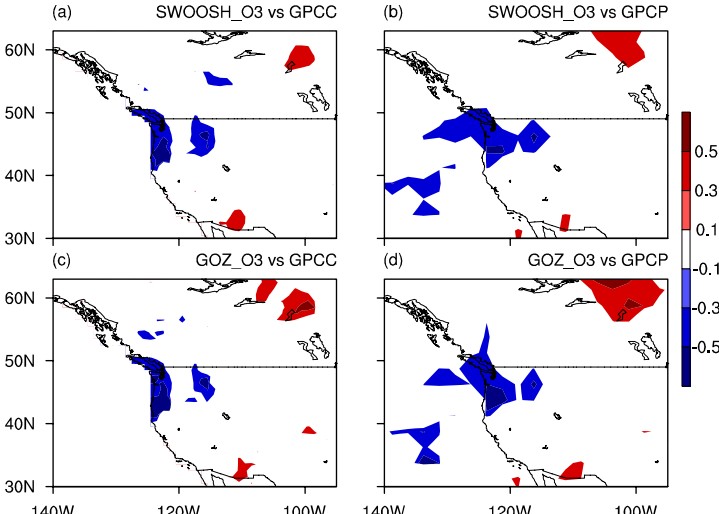


**Figure 1.** Correlation coefficients between March ASO and April precipitation

variations calculated from SWOOSH (a, b) and GOZCARDS (c, d) ozone, and GPCC

(a, c) and GPCP (b, d) rainfall for the period 1984–2016. Only regions above the 90%
confidence level are colored. The long-term linear trend and seasonal cycle in all
variables were removed before the correlation analysis.





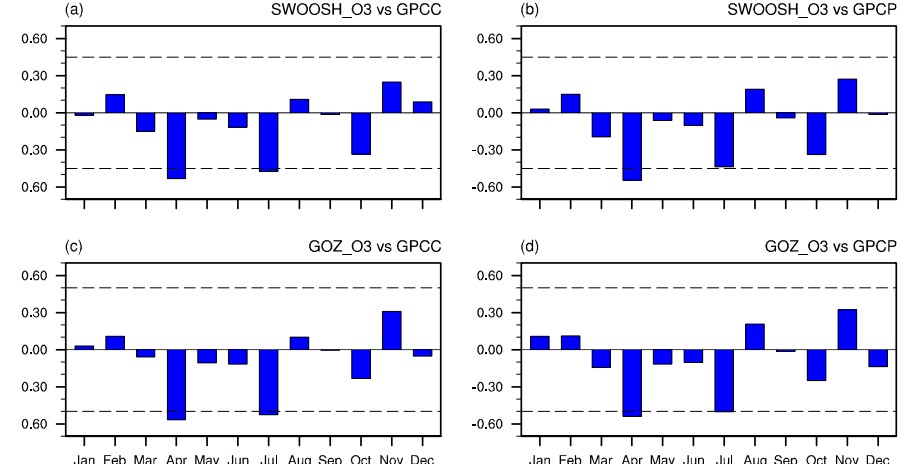

**Figure 2.** (a) Correlation coefficients between March ASO index and precipitation anomalies in the northwestern US (43°–50°N, 115°–130°W) for each month calculated from SWOOSH (a, b) and GOZCARDS (c, d) ozone, and GPCC (a, c) and GPCP (b, d) rainfall for the period 1984–2016. The long-term linear trend and seasonal cycle were removed from the original datasets before calculating the correlation coefficients.



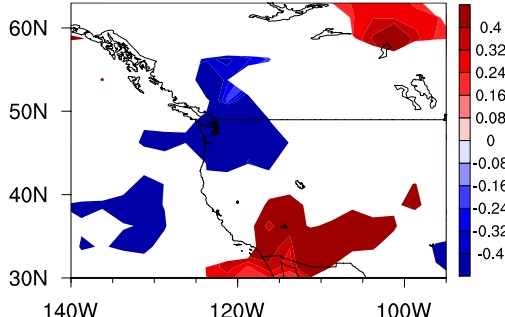

**Figure 3.** Differences in composite April precipitation (mm/day, from GPCP) anomalies in the US between positive and negative ASO anomaly events (from SWOOSH data) for 1984–2016. Only regions above the 90% confidence level are colored. See Table 2 for the definition of positive and negative March ASO anomaly events for composite analysis. Before performing the composite analysis, the seasonal cycle and linear trend were removed from the original precipitation dataset.




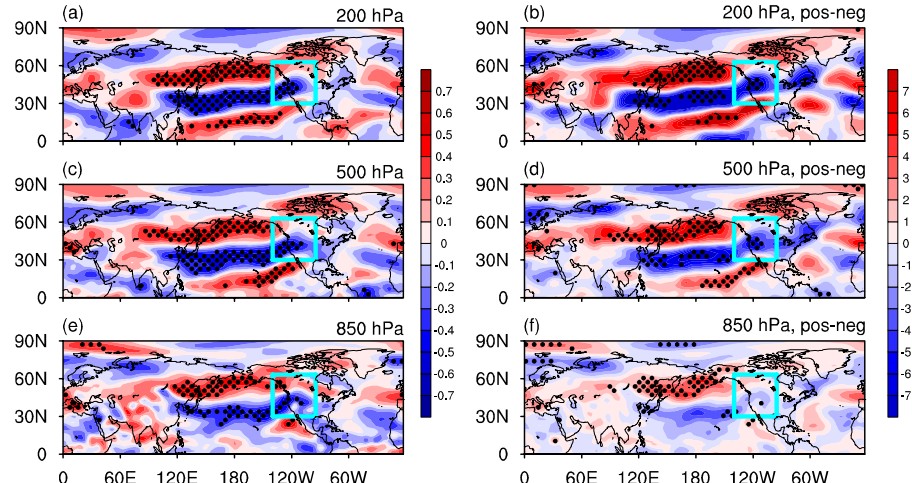

**Figure 4.** Correlation coefficients between March ASO index and April zonal wind

variations (m/s, from NCEP2) from 1984 to 2016 at 200 hPa (a), 500 hPa (c), and 850

hPa (e). Differences in composite April zonal wind (m/s) anomalies between positive

and negative ASO anomaly events are shown at 200 hPa (b), 500 hPa (d), and 850 hPa

(f). Dots denote significance at the 90% confidence level, according to Student's *t*-test.

Blue square is the area shown in Fig. 1. Before performing the analysis, the seasonal

cycle and linear trend were removed from the original datasets. Selected ASO

anomalous events are based on Table 2.





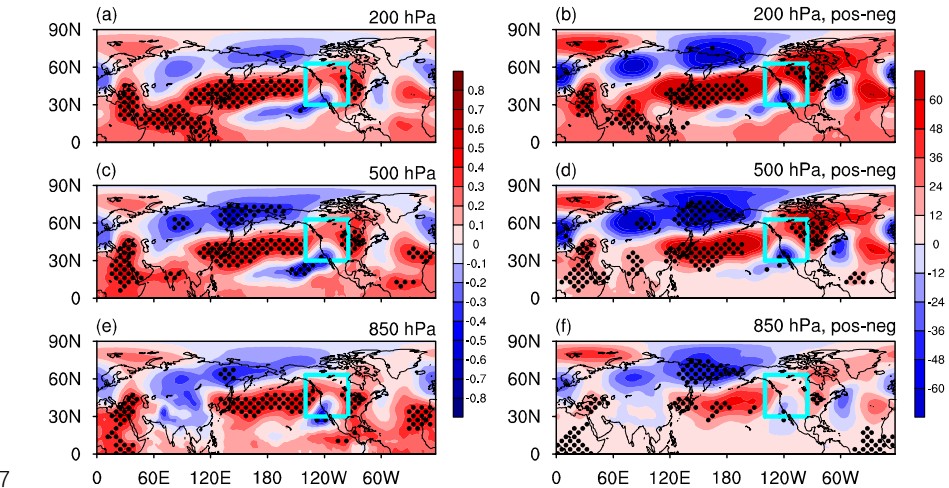

Figure 5. Same as Fig. 4, but for geopotential height (m).




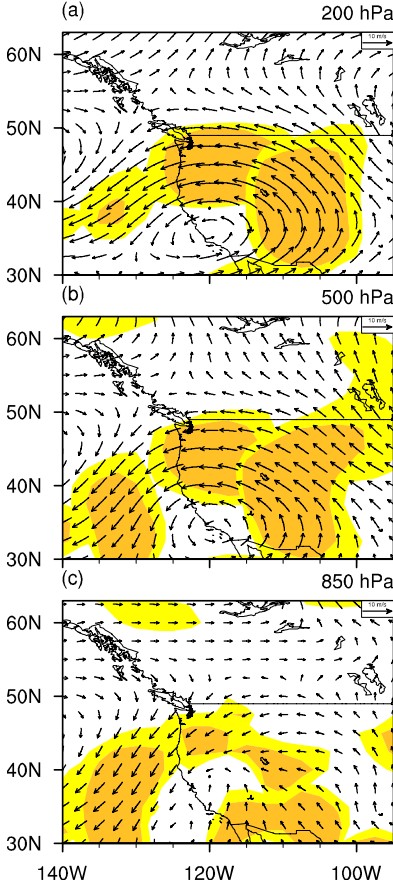

**Figure 6.** Differences in composite April winds (vectors, m/s, from NCEP2) between

positive and negative ASO anomaly events at 200 hPa (a), 500 hPa (b), and 850 hPa

(c) for 1984–2016. Colored regions are statistically significant at the 90% (light

yellow) and 95% (dark yellow) confidence levels. The seasonal cycle and linear trend

were removed from the original dataset. The ASO anomaly events are selected based

on Table 2.



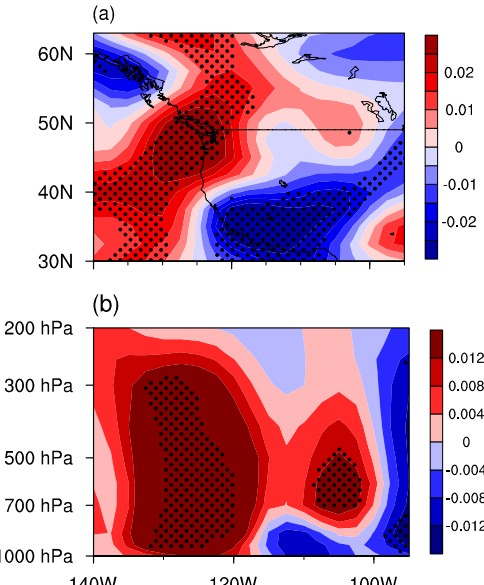

**Figure 7.** (a) Longitude–latitude cross-section of differences in composite April

vertical velocity anomalies (averaged over 1000–500 hPa) between positive and

negative ASO anomaly events for 1984–2016. (b) Longitude–height cross-section of

differences in composite April vertical velocity anomalies (averaged over 43°–50°N)

between positive and negative ASO anomaly events from 1984 to 2016. Dots denote

significance at the 90% confidence level. Before performing the analysis, the seasonal

cycle and linear trend were removed from the original dataset. The ASO anomaly

events are selected based on Table 2. The vertical velocity (Pa/s) dataset is from

NCEP2.




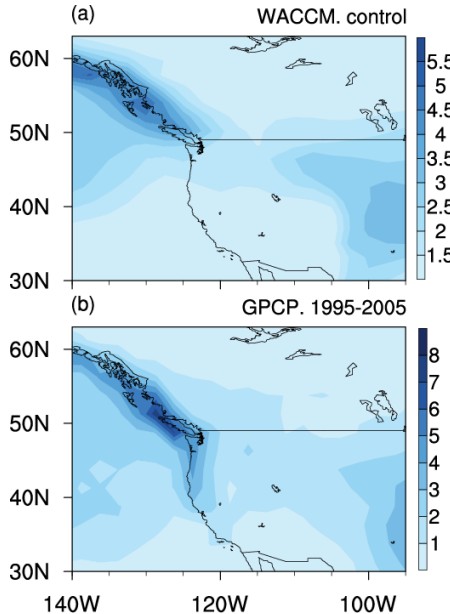

**Figure 8.** (a) Spatial distribution of April precipitation (mm/day) climatology in the

control experiment (R1). (b) Same as (a), but precipitation from the GPCP for the

period 1995–2005. For details of specific experiments, see Table 1.



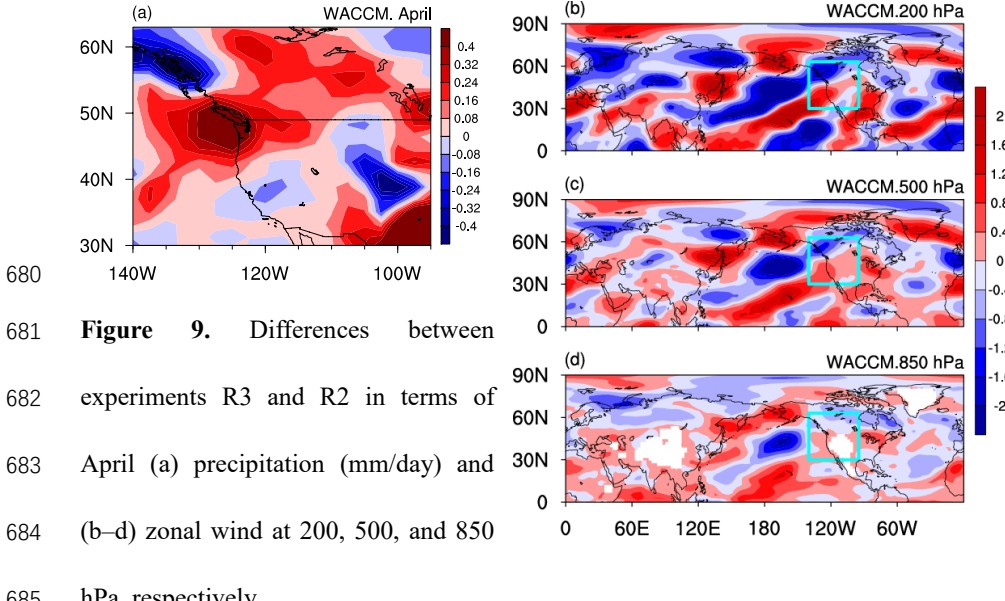

**Figure 9.** Differences between experiments R3 and R2 in terms of April (a) precipitation (mm/day) and (b–d) zonal wind at 200, 500, and 850 hPa, respectively.





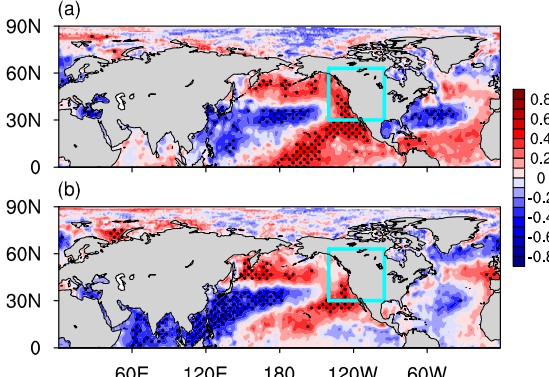

**Figure 10.** (a) Correlation coefficients between regional precipitation (43°–50°N, 115°–130°W) and SST variations in April for 1984–2016. (b) Correlation coefficients between March ASO (× –1) and April SST variations for 1984–2016. Dots denote significance at the 90% confidence level, according to Student's $t$-test. Before performing the analysis, the seasonal cycle and linear trend were removed from the original data. ASO data are from SWOOSH, precipitation from NCEP2, and SST from HadSST.





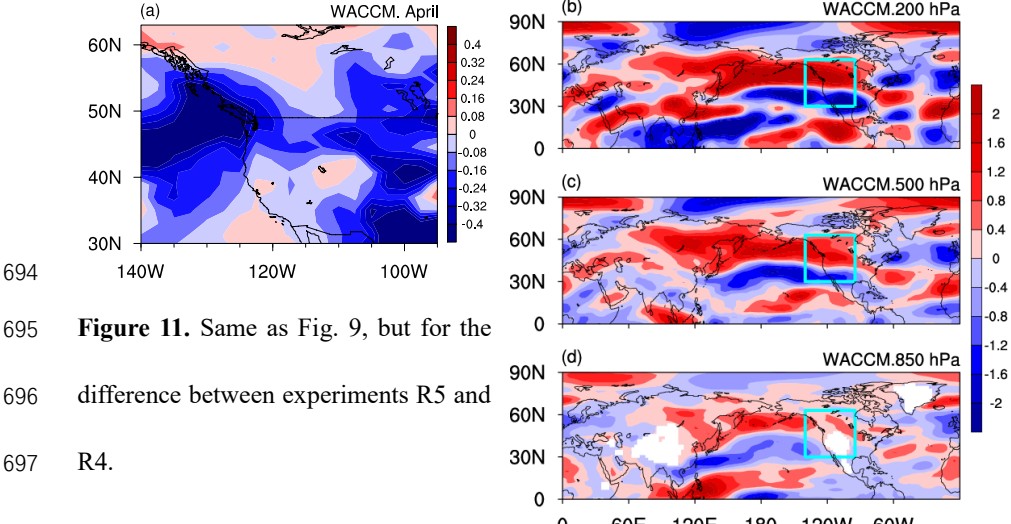

**Figure 11.** Same as Fig. 9, but for the

difference between experiments R5 and

R4.





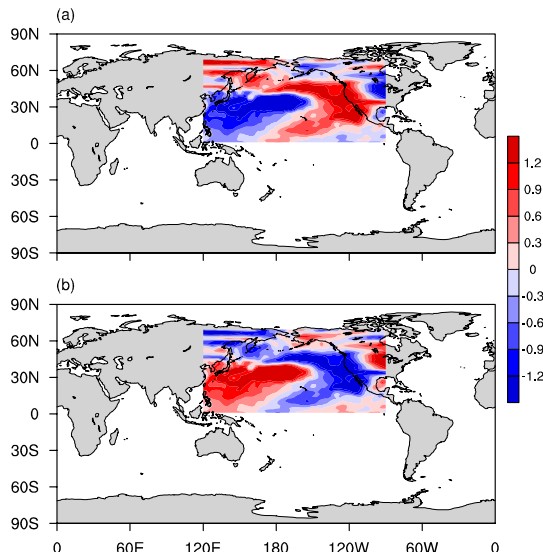

Figure 12. (a) Composite SST anomalies during negative ASO anomaly events. (b)

Composite SST anomalies during positive ASO anomaly events. The ASO anomaly

events are selected based on Table 2. SST data are from CESM SST forcing data.





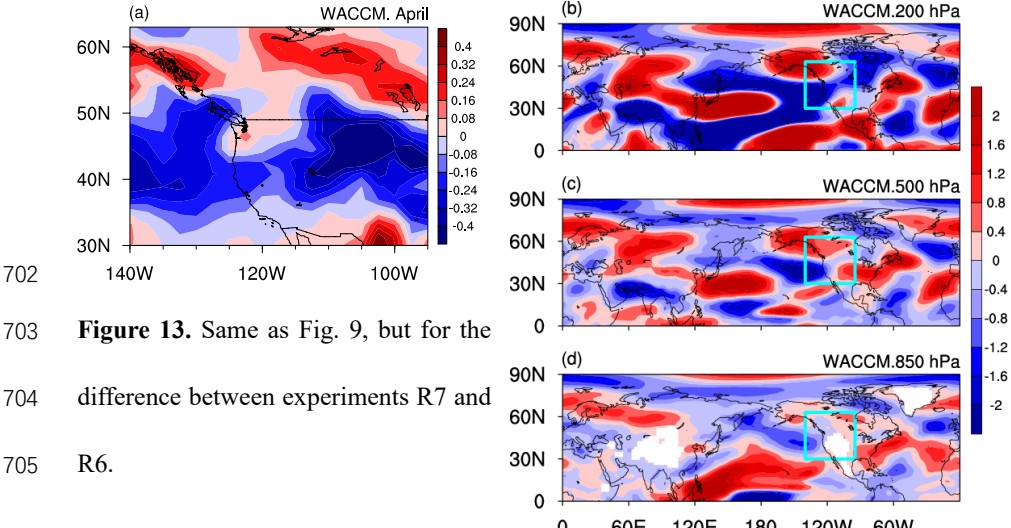

**Figure 13.** Same as Fig. 9, but for the difference between experiments R7 and R6.