# Peer review of "Effects of Arctic stratospheric ozone changes on spring precipitation in the northwestern United States"

_Atmospheric Chemistry and Physics, 2018_

## Short Comment (SC1) · 8 Jun 2018

A highly interesting study - just one question for clarification:

You decrease/increase the ozone climatology homogeneously by 15% in R2/R3, which will also amplify zonal inhomogeneity in the ozone climatology because already greater ozone mixing ratios will be increased more in terms of absolute magnitude. Several studies (e.g. Gabriel et al. 2007, Gillet et al. 2009, McCormack et al. 2011, Nowack et al. 2018) showed that such zonal asymmetry can be important for the Arctic vortex climatology and as a result surface climate. Do you have any means of determining the importance of the general increase/decrease in ozone imposed by you as compared

to the amplification of the zonal structure, which might be particularly important for the vortex climatology? It would be great if you could put your results into context.

McCormack, J. P., Nathan, T. R. & Cordero, E. C. (2011), 'The effect of zonally asymmetric ozone heating on the Northern Hemisphere winter polar stratosphere', Geophysical Research Letters 38(3), 1–5.

Gabriel, A., Peters, D., Kirchner, I. & Graf, H. F. (2007), 'Effect of zonally asymmetric ozone on stratospheric temperature and planetary wave propagation', Geophysical Research Letters 34(6).

Gillett, N. P., Scinocca, J. F., Plummer, D. A. & Reader, M. C. (2009), 'Sensitivity of climate to dynamically-consistent zonal asymmetries in ozone', Geophysical Research Letters 36(10), 1–5.

Nowack, P. J., Abraham, N. L., Braesicke, P. & Pyle, J. A. (2018), 'The impact of stratospheric ozone feedbacks on climate sensitivity estimates', Journal of Geophysical Research: Atmospheres 123, 4630-4641.
* * *

---

## Short Comment (SC2) · 28 Jun 2018

**Responses to Short Comments**

**Effects of Arctic stratospheric ozone changes on spring precipitation in the northwestern United States (ACP-2018-414)**

Xuan Ma, Fei Xie, Jianping Li, Wenshou Tian, Ruiqiang Ding, Cheng Sun, and Jiankai Zhang

June 2018

*You decrease/increase the ozone climatology homogeneously by 15% in R2/R3, which will also amplify zonal inhomogeneity in the ozone climatology because already greater ozone mixing ratios will be increased more in terms of absolute magnitude. Several studies (e.g. Gabriel et al. 2007, Gillet et al. 2009, McCormack et al. 2011, Nowack et al. 2018) showed that such zonal asymmetry can be important for the Arctic vortex climatology and as a result surface climate. Do you have any means of determining the importance of the general increase/decrease in ozone imposed by you as compared to the amplification of the zonal structure, which might be particularly important for the vortex climatology? It would be great if you could put your results into context.*

**Response: We thank the reviewer for the positive evaluation of our study and sincerely appreciate the reviewer's insightful and helpful comments. We are also sorry for missing some important references in the manuscript. The following references had been added in the revised manuscript (*Gabriel et al., 2007; Gillett et al., 2009; Nowack et al., 2015, 2017, 2018; McCormack et al., 2011*).**

**The experiments preformed in this study are in order to confirm that whether the observed positive/negative ASO anomalies could force the abnormal precipitation in the northwestern United States. Fig. R1a and b show the observed ozone anomalies during negative and positive ASO anomalies events, respectively. Fig. R1c and d show the negative ozone forcing in R2 and positive ozone forcing in R3, respectively. Fig. R1 illustrates that the ozone forcings imposed in R2/R3 is similar to the observations, indicating the ozone forcings given in the experiments are reasonable. The results in our manuscript show that this kind of ozone forcing indeed could cause the observed precipitation anomalies in the northwestern United States.**

**The question, i.e., "*general increase/decrease in ozone or the amplification of zonal structure of ozone, which might be particularly important for the vortex climatology*" is a very important question; however, we think it isn't the focus of this article. This issue needs a further proof and a lot of experiments to demonstrate. If we put these contents in the text, it will make the manuscript too**

**long and too complicated. This point is a good idea, and we will try to finish it in the next work.**

[Figure]

**Figure R1. The composite ozone anomalies (ppmv) during negative (a) and positive (b) ASO anomalies events, based on SWOOSH ozone data from 1984 to 2016. The definition of ASO anomalies events please refers to manuscript. The negative ozone forcing in R2 (c) and positive ozone forcing in R3 (d) in WACCM4 experiments, respectively.**

**References:**

Gabriel, A., Peters, D., Kirchner, I. & Graf, H. F. (2007), 'Effect of zonally asymmetric ozone on stratospheric temperature and planetary wave propagation', *Geophysical Research Letters 34(6).*

Gillett, N. P., Scinocca, J. F., Plummer, D. A. & Reader, M. C. (2009), 'Sensitivity of climate to dynamically-consistent zonal asymmetries in ozone', *Geophysical Research Letters 36(10), 1–5.*

Nowack, P. J., Abraham, N. L., Braesicke, P. & Pyle, J. A. (2018), 'The impact of stratospheric ozone feedbacks on climate sensitivity estimates', *Journal of Geophysical Research: Atmospheres 123, 4630-4641.*

Nowack, P.J., Braesicke, P., Abraham, N. L. & Pyle, J. A. (2017), 'On the role of ozone feedback in the ENSO amplitude response under global warming', *Geophysical Research Letters 44, 3858–3866.*

Nowack, P. J., Abraham, N. L., Maycock, A. C., Braesicke, P., Gregory, J. M., Joshi, M. M.,

*Osprey, A. & Pyle, J. A. (2015). A large ozone-circulation feedback and its implications for global warming assessments. Nature climate change, 5, 41-45.*

*McCormack, J. P., Nathan, T. R. & Cordero, E. C. (2011), 'The effect of zonally asymmetric ozone heating on the Northern Hemisphere winter polar stratosphere', Geophysical Research Letters 38(3), 1–5.*

---

## Referee Comment (RC1) · Anonymous Referee #1 · 30 Jun 2018

This study examines the relationship between springtime Arctic lower stratospheric ozone concentrations and precipitation anomalies over the northwestern United States (Washington and Oregon). Using observations and WACCM model simulations (with various prescriptions of ozone and SSTs), the authors link Arctic lower stratospheric ozone depletion to precipitation increases over the northwestern United States. Their model simulations indicate that prescribing both the ozone and SSTs is necessary to recover the observed relationship in the model.

The premise of this study is very interesting . . . using Arctic lower stratospheric ozone anomalies to predict springtime precipitation. However, as written, I don't find the

manuscript to meet the standards of an ACP publication for the following reasons: 1) most of the observed correlations are based upon statistical significance at the 90% confidence level, 2) the authors fail to account for the role of stratospheric dynamic variability (sudden stratospheric warmings) in their analysis, 3) many of the figures (and associated text) simply repeat the same information, and most importantly 4) no physical mechanism is provided to explain why lower stratospheric ozone anomalies impact the North Pacific circulation (but not the North Atlantic circulation) and in particular how they can excite SST anomalies (which seem opposite to those that would be forced by the lower tropospheric wind anomalies). For these reasons, I am inclined to recommend that the paper be rejected at this point and encourage the authors to resubmit their interesting analyses after they have addressed some of these issues.

Major Revisions

1. Winters with sudden stratospheric warmings and strong stratospheric polar vortices are caused by natural wave-driven dynamic variability (lines 67-68), and thus chemical ozone depletion will only occur when the Arctic stratosphere is not dynamically active (strong stratospheric polar vortex years). So, Arctic stratospheric ozone (ASO) depletion is only relevant in years when the dynamics precondition the Arctic stratosphere for it to potentially occur. It's not immediately apparent to me what advantage looking at ozone (compared to polar stratospheric temperature anomalies) provides for tropospheric teleconnections. In other words, if instead of using ozone as a criteria for the years selected in Table 2, you used the strength of the stratospheric polar vortex, would you get the same patterns? Or, another way of stating this, are the years with positive ASO anomalies associated with sudden stratospheric warmings and/or early seasonal breakdowns of the stratospheric polar vortex? The paper is framed as if ozone is the predominant cause of NH stratospheric circulation anomalies. In reality, the ozone-induced stratosphere-troposphere connections should be secondary in importance to those driven by stratospheric dynamics in the NH. In the SH, where year-to-year dynamic variability is weaker, the ozone-induced stratosphere-troposphere connections

are more prominent.

2. The authors state that ASO recovery will cause the northwestern United States to become drier in the future (lines 19–20, lines 203–205). The analysis in this study is based entirely on detrended ozone anomalies (year-to-year variability). If the authors wish to make this argument, they will need to convincingly show that 1) springtime ASO has trended downward in recent years and 2) northwestern US precipitation has trended upward during April over the same time interval (independent of concurrent variability in ENSO and the PDO).

3. Prior studies have argued that stratospheric circulation anomalies can couple down into the troposphere with a spatial pattern similar to the Northern Annular Mode (NAM) or North Atlantic Oscillation (NAO) (lines 70-71). Yet, the authors' analysis shows a poleward circulation shift over the North Pacific, but not the North Atlantic. Some discussion needs to be provided about why the authors' results are different than those documented in previous studies. It would be nice to compare the patterns shown in Figs. 3–7 with those associated with the NAM/NAO. Additionally, given that the SST anomalies shown in Fig. 10 strongly resemble the Pacific Decadal Oscillation (PDO), the same patterns should be examined for the PDO and ENSO. With such a small sample size of years used in the analysis (Table 2), the authors could simply be sampling concurrent SST variability. Given that most previous studies on this subject see the strongest anomalies in the North Atlantic sector, the fact that all of the anomalies are in the Pacific in this study makes me concerned that Pacific SST variability is being aliased into the analysis.

4. As a related point, variations in March ASO should be linked closely to the timing of the seasonal breakdown of the NH stratospheric polar vortex. Black has examined this issue in detail in a series of papers (e.g., Black et al. 2006). Again, the authors need to better contextualize their results in the context of the past literature, which emphasizes the North Atlantic.

5. A consistent measure of statistical significance needs to be provided throughout the paper. Some figures show a 90% level, others a 95% level, and some show no significance at all (model results). 90% is a fairly weak threshold for statistical significance (1 in 10 chance that the point is significant by chance). I would recommend using the 95% level, or at least showing both the 90% and 95% levels (as is shown in Fig. 6).

6. Related to point #3 above, how can we be sure that the SST anomalies in Fig. 10 are in fact caused by the stratospheric anomalies? They seem inconsistent with the wind anomalies in Fig. 4 (enhanced air-sea fluxes and cooling should occur in regions of enhanced westerlies). Some physical mechanism linking ASO to the SST anomalies needs to be provided. Without prescribing ASO anomalies in a fully coupled model (with interactive SSTs), it's difficult to conclusively establish that the SST anomalies can in fact be forced by ASO.

Minor Revisions

7. The following sentence structure used in the abstract (and elsewhere in the paper) is very difficult to read:

"An increase (decrease) …. results in enhanced (weakened) … but weakened (enhanced) … facilitating (impeding) …"

Please consider eliminating the words in parentheses, or using a difficult format to convey this information. It's confusing to discuss both polarities (both an increase and decrease in ozone) within the same sentence structure.

8. Line 37: The circulation changes mostly occurred in the late 20th century, not the early 21st century, as the ozone hole was increasing in size from the 1980s until around the year 2000. Since that time, the ozone hole has stabilized in size, and may in fact be starting to recover (see Solomon et al. 2017).

9. Lines 44–45. See Fig. 3 in Kang et al. (2011). The precipitation changes associated with Antarctic stratospheric ozone depletion are more accurately described as an

increase in the subtropics and high latitudes, and a decrease at mid-latitudes.

10. Lines 55–57: This explanation of the surface temperature anomalies associated with Antarctic ozone depletion is not consistent with previous literature. See discussion in Thompson et al. (2011) and the references therein. The surface temperature anomalies are linked to how the poleward circulation shift associated with the ozone hole affects localized wind patterns (and associated thermal advection) at each location.

11. Line 110: The vertical pressure range (100–50 hPa) contradicts that in footnote #2 of Table 1 (150–50 hPa). Please correct.

12. Line 142: If SSTs are specified, the term "coupled" here is misleading. Follow convention in the literature, I would recommend using the term "coupled" only if the atmosphere model is fully coupled to an interactive ocean model.

13. Line 148: This statement seems to contradict the statement on line 143. The model has middle atmospheric chemistry, yet the model does not include interactive chemistry. This needs to be clarified.

14. Lines 151, 154: The text refers to a reference period of 1980–2015, while Table 1 refers to a reference period of 1995–2005. This needs to be clarified and standardized throughout the paper.

15. Line 164: I would use "break down" rather than "rupture" here to be consistent with terminology in previous literature.

16. Line 167: This lead time is not unique to NH stratospheric ozone perturbations. It is consistent with the tropospheric anomalies associated with NH sudden stratospheric warmings (Baldwin and Dunkerton 2001) and SH stratospheric ozone depletion (Thompson and Solomon 2002).

17. Lines 236–240: This statement is not consistent with the figures. Figures 4–6 show a barotropic circulation response (same sign throughout the depth of the troposphere),

with an anomalous cyclone over western North America at all levels (Fig. 6).

18. Lines 248–250: How so? I don't understand the dynamical basis for this statement.

19. Lines 254, 258, 365: Reanalyses cannot adequately resolve convective activity. The anomalous downwelling here is associated with synoptic-scale processes (see positive geopotential heights in northeast Pacific in Fig. 5). The pattern in Fig. 7a should closely correspond to sea-level pressure anomalies (a surface high in the northwestern United States and a surface low in the southwestern United States).

20. Line 282: It doesn't look like opposite to me ... just shifted a little further to the north in the model than in the observations (which, of course, would make a difference for regional impacts as the authors nicely state on the subsequent lines).

21. Line 301: The SST pattern looks a lot like the Pacific Decadal Oscillation (PDO) or the North Pacific Mode (Hartmann 2015). How well correlated is the time series of the "Victoria Mode" with these modes?

22. All figures: I think it's unnecessary to show both the correlations and composite differences (Figs. 1 and 3, left and right columns of Fig. 4-5), as they basically convey identical information.

23. Figures 1 and 3: It's difficult to interpret these patterns with so much of the map left blank. I would recommend showing the correlation coefficients for the entire map, and stippling those regions that are statistically significant.

24. Figure 2: What are the dashed black lines? A measure of statistical significance?

25. Figures 4–5: Is it necessary to show both geopotential heights and zonal wind? Both figures convey exactly the same information (via geostrophic balance).

26. Figure 7: It would be good to clarify that blue is upward motion and red is downward motion.

27. Figure 12: Because the model has prescribed SSTs, how do you know the model

SSTs associated with ASO anomalies? Are these some version of the observed SST anomalies as they don't look exactly like those in Fig. 10?

Typos

Line 96: central of China -> central China

Line 134: regarding -> regarded

Line 147: is at -> are at

Lines 173, 354: Washington and Oregon states -> Washington and Oregon

Line 176: the Fig. 1 -> Fig. 1

Line 259: enhances -> weakens

Table 1, R4–R7: a SST anomalies -> SST anomalies

References

Black, R.X., B.A. McDaniel, and W.A. Robinson, 2006: Stratosphere–Troposphere Coupling during Spring Onset. J. Climate, 19, 4891–4901, https://doi.org/10.1175/JCLI3907.1

Hartmann, D. L. (2015), Pacific sea surface temperature and the winter of 2014. Geophys. Res. Lett., 42, 1894–1902. doi: 10.1002/2015GL063083.

Solomon, S., D. Ivy, M. Gupta, J. Bandoro, B. Santer, Q. Fu, P. Lin, R. R. Garcia, D. Kinnison, and M. Mills, 2017: Mirrored changes in Antarctic ozone and stratospheric temperature in the late 20th versus early 21st centuries. J. Geophys. Res. Atmos., 122, 8940–8950, doi:10.1002/2017JD026719.

---

## Referee Comment (RC2) · Anonymous Referee #2 · 5 Jul 2018

The manuscript presents a well-designed study of the effects of variability in spring-time Arctic stratospheric ozone (ASO) on the tropospheric circulation over the Pacific basin, extending into the north-west United States. The authors present statistical relationships between a variety of physical climate variables and ASO in observations, finding an inverse correlation between ASO anomalies and March precipitation over the north-west United States, and then explore the causality with a number of WACCM model simulations using anomalies applied to the prescribed ozone and sea-surface temperatures used in the model. The model simulations provide convincing evidence that the combined effect of the ASO anomalies and correlated changes in sea-surface temperatures over the Pacific can reproduce the observed pattern of changes in winds

and precipitation. The study is well thought out and presented and I have no serious concerns about the methodology. The one significant missing aspect to the manuscript is the way the authors discuss ozone variability and the effects of ozone variability on dynamics as a completely independent forcing. The model simulations are convincing in that the specified ozone can be modified and the impact on the dynamics can be estimated in a one-way cause-and-effect manner. But in the real atmosphere there is a very tight coupling between dynamical modes of variability and Arctic ozone. Variability in the amount of planetary wave forcing from the troposphere has a direct connection to the strength of the Brewer-Dobson circulation and the amount of poleward ozone transport each year. And the occurrence of Sudden Stratospheric Warmings in the late winter or early spring can determine whether polar stratospheric temperatures cold enough for heterogenous chemistry on polar stratospheric clouds will occur and produce significant chemical ozone destruction in the Arctic. I think there are two important implications for the manuscript under consideration here. One is that the observation-based analysis must discuss the strong coupling between dynamical variability and ozone variability and must recognize that the correlations of certain physical variables with ozone also reflect correlations with other aspects of dynamical variability. And second, I believe the authors cannot state that the Victoria Mode anomalies in Pacific sea-surface temperatures are caused by, as opposed to being associated with, the ASO anomalies.

As given below in the minor comments, in a few places through the manuscript the differences in the circulation between different WACCM experiments are described in very direct ways. It would be much more illustrative for the reader if these changes could be associated with changes in the position of significant climatological features, in a similar way that the Antarctic wind changes can be summarized as a pole-ward shift of the jet.

Minor comments:

Lines 15 – 18: Following my concerns about correlation and causality, the sentence 'In

addition, the ASO changes cause sea surface temperature anomalies over the North Pacific that would cooperate with the ASO changes to modify the circulation anomalies over the northwestern US.' should be softened.

Lines 109 - 111: As stated here, the ASO is calculated as an anomaly after removing the annual cycle and trend. I would imagine the long-term trend is predominately due to the rise in ozone depleting substances. Why was the trend removed from the calculation of ASO, as I would think the March ASO anomaly related to ozone depletion would be part of the signal you are looking for? And is the trend calculated as a single linear trend across the entire period or some measure that is related to halogen loading in the stratosphere such as Equivalent Effective Stratospheric Chlorine (EESC)? As the period analysed is 1984 – 2015, or so, this would include both the rapid increase in EESC up to ∼2000 and the plateau or slow decline since then and a single linear trend across the entire period would be a less than ideal estimate of the forced response.

Line 117: 'Another set of ozone dataset is...' sounds a bit redundant. Could I suggest 'Another set of ozone data is...'

Lines 149 – 151: The statement 'The model's radiation scheme uses these conditions: fixed greenhouse gas (GHG) values, averages of emissions scenario A2 of the Intergovernmental Panel on Climate Change (IPCC) (WMO, 2003) for 1980–2015.' is difficult to interpret. Is it that the fixed GHG values that were used are the 1980-2015 average from the A2 scenario? It seems a bit clearer in the text in Table 1, but there the average is said to be over 1995-2005.

Lines 212 – 214: The correlation of zonal wind anomalies with the ASO is described as: 'This implies that the increase (decrease) in ASO can result in enhanced (weakened) westerlies in the high and low latitudes of the North Pacific but weakened (enhanced) westerlies in the mid-latitudes.' The changes in southern hemisphere winds associated with ozone depletion are often described in terms of a shift of the jet that produces a dipole pattern of changes in wind. Here the authors argue that the ASO is associated

with a tripole of changes in zonal wind. Do the authors have an explanation for the pattern of changes that can be related to shifts or changes in magnitude of climatological features like the Aleutian Low? And can other explanations for the changes at low latitudes, such as ENSO, be ruled out?

Lines 236 – 240: 'This kind of circulation anomaly corresponds to an anomalous cyclone (anticyclone) in the western US in the middle and upper troposphere, which is likely associated with a strong low (high) pressure system in the middle and upper troposphere and a relatively weak high (low) pressure system in the lower troposphere.' I can see how this description fits with the pattern of wind changes shown in Figure 6, but that the pattern of changes shown in panel (A), for example, showing a cyclonic pattern centered over the south-western US does not necessarily mean that this is caused by the appearance of a well-defined, anomalous cyclone. While the pattern of the differences is cyclonic, it could be due to the weakening of an anticyclone? The description would have a stronger physical basis if the changes were related to changes in the strength of position of well-recognized climatological features.

Lines 248 -250: 'In addition, a strong low-pressure system in the middle and upper troposphere over the western US during positive ASO anomaly events (Fig. 6) suggests downwelling flow in the region.' Similar to the concerns about the interpretation of Lines 236 – 240, there is a direct link made between a pattern of changes and the appearance of a particular meteorological feature.

Lines 251 – 262: While I can understand how changes in vertical velocity (w) are coherent with the large-scale changes in circulation, the text in this paragraph makes a direct link between changes in w from the NCEP2 reanalysis and changes in convective precipitation. For example, at lines 253 – 255: 'When the March ASO increases, tropospheric convective activity in the northwestern US (115°–130° W) weakens, corresponding to anomalous downwelling.' Can a direct link between convective precipitation and changes in monthly-average vertical velocity be made? I think the authors would need to support this statement with citations to previous work. I am also somewhat sceptical about the general direction of the argument, which appears to be trying to link the circulation changes to precipitation changes. Is convective precipitation an important fraction of precipitation in the north-west US in March-April? I would have thought the precipitation changes shown in Figure 1 are a much more straight-forward reflection of changes in orographic precipitation related to the decrease in wind and (presumably) moisture transport?

Lines 267 – 268: The WACCM experiments detailed in Table 1 show that the perturbed ASO simulations vary ozone by +/- 15% between 30N and 90N. How realistic is this perturbation compared with the estimates from SWOOSH and GOZCARDS datasets? Perhaps a figure of the zonal-average difference could be included for the composite positive and negative ASO years? At high latitudes a +/-15% variability does not sound too large, perhaps even a bit small, but a +/- 15% change at 30N seems quite large.

Line 275: Beginning here, the results from the WACCM simulations are presented. Figures 9, 11 and 13, which show the differences between the WACCM experiments do not have any indication of the statistical significance. All of the other difference plots did have some manner of denoting statistical significance at the 90% level and these three plots should as well.

---

## Author Comment (AC1) · 13 Sep 2018

**Responses to Referees' Comments**

**Effects of Arctic stratospheric ozone changes on spring precipitation in the northwestern United States (ACP-2018-414)**

Xuan Ma, Fei Xie, Jianping Li, Xinlong Zheng, Wenshou Tian,
Ruiqiang Ding, Cheng Sun, and Jiankai Zhang

September 2018

**Response to Referee 1**

*This study examines the relationship between springtime Arctic lower stratospheric ozone concentrations and precipitation anomalies over the northwestern United States (Washington and Oregon). Using observations and WACCM model simulations (with various prescriptions of ozone and SSTs), the authors link Arctic lower stratospheric ozone depletion to precipitation increases over the northwestern United States. Their model simulations indicate that prescribing both the ozone and SSTs is necessary to recover the observed relationship in the model.*

*The premise of this study is very interesting ... using Arctic lower stratospheric ozone anomalies to predict springtime precipitation. However, as written, I don't find the manuscript to meet the standards of an ACP publication for the following reasons: 1) most of the observed correlations are based upon statistical significance at the 90% confidence level, 2) the authors fail to account for the role of stratospheric dynamic variability (sudden stratospheric warmings) in their analysis, 3) many of the figures (and associated text) simply repeat the same information, and most importantly 4) no physical mechanism is provided to explain why lower stratospheric ozone anomalies impact the North Pacific circulation (but not the North Atlantic circulation) and in particular how they can excite SST anomalies (which seem opposite to those that would be forced by the lower tropospheric wind anomalies). For these reasons, I am inclined to recommend that the paper be rejected at this point and encourage the authors to resubmit their interesting analyses after they have addressed some of these issues.*

**Response: We thank the reviewer for taking the time to assess the manuscript and for highlighting important issues and providing helpful comments and suggestions to improve the manuscript. We have revised the manuscript carefully according to the reviewer's comments. The 95% confidence level is now used throughout the paper. Some less-important figures have been removed. We apologize for the lack of clarity in places, which led the reviewer to feel that the manuscript overemphasized the influence of ozone on stratosphere and troposphere coupling.**

At the same time, we are grateful for the important references provided by the reviewers; the new references, including Black et al. (2005, 2006, 2009); Gabriel et al. (2007); Gillett et al. (2009); Nowack et al. (2015, 2017, 2018); McCormack et al. (2011); WMO (2003) and Zhang et al. (2018), have been cited in the revised manuscript. Please see the following detailed point-by-point responses:

*Major Comments:*

*1. Winters with sudden stratospheric warmings and strong stratospheric polar vortices are caused by natusral wave-driven dynamic variability (lines 67-68), and thus chemical ozone depletion will only occur when the Arctic stratosphere is not dynamically active (strong stratospheric polar vortex years). So, Arctic stratospheric ozone (ASO) depletion is only relevant in years when the dynamics precondition the Arctic stratosphere for it to potentially occur. It's not immediately apparent to me what advantage looking at ozone (compared to polar stratospheric temperature anomalies) provides for tropospheric teleconnections. In other words, if instead of using ozone as a criteria for the years selected in Table 2, you used the strength of the stratospheric polar vortex, would you get the same patterns? Or, another way of stating this, are the years with positive ASO anomalies associated with sudden stratospheric warmings and/or early seasonal breakdowns of the stratospheric polar vortex? The paper is framed as if ozone is the predominant cause of NH stratospheric circulation anomalies. In reality, the ozone-induced stratosphere-troposphere connections should be secondary in importance to those driven by stratospheric dynamics in the NH. In the SH, where year-to-year dynamic variability is weaker, the ozone-induced stratosphere-troposphere connections.*

**Response: Thanks for the comment. We agree with the reviewer's opinion that the spring ASO variations are related to changes in the winter Arctic stratospheric vortex (SPV). The strength of the SPV can affect ASO, and then ASO affects tropospheric teleconnection and precipitation in the northwestern United States (indirect effect of SPV). The strength of the SPV may also have a direct leading effect on tropospheric teleconnection and precipitation in the northwestern United**

States. There is a tight coupling between dynamical modes of variability and ASO. In this study, we have not thought of a better way to separate the two effects on precipitation. Thus, the thrust of this study is to at least recognize that the ASO changes may affect precipitation in the northwestern United States. From the analysis of observational data, we find that the ASO has a leading relationship with spring precipitation in the northwestern United States. In addition, this relationship can be reproduced in simulations by abnormal ASO forcing. This implies that the variations in spring ASO can force the observed tropospheric circulation and precipitation anomalies in the northwestern United States.

Figure R1 shows the correlation coefficients between the February SPV (multiplied by –1) index and April 200 hPa zonal wind and precipitation variations (Fig. R1a and b), and between March ASO and April 200 hPa zonal wind and precipitation (Fig. R1c and d). The SPV index is defined as the strength of the stratospheric polar vortex, following Zhang et al. (2018). Although the patterns of correlation coefficients in Fig. R1 are similar, the ASO variations are much closer than the SPV to the variations in 200 hPa zonal wind and precipitation. Fig. R1 indicates indirect and direct effects of winter SPV on spring tropospheric climate. Since the coupling between dynamical and radiative processes in spring is strong, the connection between winter SPV and spring tropospheric circulation seems weaker than that between the spring ASO and tropospheric circulation.

[Figure]

**Figure R1**. (a) Correlation coefficients between the February –SPV ($10^5$ K m$^2$ kg$^{-1}$ s$^{-1}$) index defined by Zhang et al. (2018) and April zonal wind variations at 200 hPa for 1984–2016. (b) Correlation coefficients between February –SPV index and April precipitation variations. (c) and (d) As for (a) and (b), but between March ASO and April 200 hPa zonal wind and April precipitation variations. Dots denote significance at the 95% confidence level, according to Student's *t*-test. The long-term linear trend and seasonal cycle in all variables were removed before the correlation analysis. The ASO data are from SWOOSH, zonal wind from NCEP2, and precipitation from GPCP.

We apologize for the lack of clarity that led the reviewer to feel that the manuscript overemphasized the influence of ozone on stratosphere and troposphere coupling in spring. In this study, we want to state that the ASO changes possibly influence precipitation in the northwestern United States, emphasizing the influence of stratospheric ozone on tropospheric regional climate. The direct and indirect impacts of SPV on precipitation in the northwestern United States and the effect of the strong coupling between dynamical and ozone variability are indeed important issues that we will examine in future work.

We have made this point clearer in the revised manuscript. The Fig. R1 and relevant discussion have been added to the discussion section in the revised manuscript. See lines 368–388.

References:

Zhang J., et al.: Stratospheric ozone loss over the Eurasian continent induced by the polar vortex shift, Nat. Commun., 9, 206, 2018.

*2. The authors state that ASO recovery will cause the northwestern United States to become drier in the future (lines 19–20, lines 203–205). The analysis in this study is based entirely on detrended ozone anomalies (year-to-year variability). If the authors wish to make this argument, they will need to convincingly show that 1) springtime ASO has trended downward in recent years and 2) northwestern US precipitation has*

*trended upward during April over the same time interval (independent of concurrent variability in ENSO and the PDO).*

**Response: Thank you for this hint. We removed this statement in the revised paper which might be too hasty. We think this issue should be a very interesting study and we will continue to work on it.**

*3.1 Prior studies have argued that stratospheric circulation anomalies can couple down into the troposphere with a spatial pattern similar to the Northern Annular Mode (NAM) or North Atlantic Oscillation (NAO) (lines 70-71). Yet, the authors' analysis shows a poleward circulation shift over the North Pacific, but not the North Atlantic. Some discussion needs to be provided about why the authors' results are different than those documented in previous studies. It would be nice to compare the patterns shown in Figs. 3–7 with those associated with the NAM/NAO.*

**Response: Thank you for this comment. In spring, stratospheric circulation anomalies, related to ASO changes, couple down into the troposphere with a spatial pattern similar to the North Pacific Oscillation (NPO). This is consistent with previous studies based on simulations (Smith et al., 2014; Calvo et al., 2015) and observations (Xie et al., 2016, 2017; Ivy et al., 2017). Stratospheric circulation anomalies that couple down into the troposphere with a spatial pattern similar to the Northern Annular Mode (NAM) or North Atlantic Oscillation (NAO) occur mainly in winter. The different pathways of stratospheric circulation anomalies from the stratosphere to troposphere may be associated with different seasons or different processes. This topic is also worthy of further study.**

**The text has been revised as follows (lines 67–71 in the revised manuscript):**

**"*Comparing with the effect of the winter stratospheric dynamical processes on the tropospheric North Atlantic Oscillation (NAO) and the incidence of extreme weather events (Baldwin and Dunkerton, 2001; Black et al., 2005, 2006, 2009), the depletion of spring ASO can cause circulation anomalies that influence the North Pacific Oscillation.*"**

**Following the reviewer's suggestion, the patterns (Figs. 3–7 in the initial**

manuscript) associated with the April NAO are shown in Figs. R2–4. Figure R2 shows that the NAO index is significantly correlated with precipitation variations in the central United States in April (not in the northwestern United States in our study). The zonal winds and geopotential height changes related to the NAO index are located mainly over the North Atlantic and North America in April (Fig. R3). An anomalous anticyclone is forced over the western United States, but the region of significant correlation is located mainly over Canada (Fig. R4).

Considering the length of the article, the number of figures, and the relevance of the content, the results in Figs. R2–4 are not included in the revised manuscript.

[Figure]

Figure R2. Correlation coefficients between NAO index and precipitation variations in April based on GPCC (a) and GPCP (b) rainfall for the period 1984–2016. Dots denote significance at the 95% confidence level, according to Student's *t*-test. The long-term linear trend and seasonal cycle in all variables were removed before the correlation analysis. The NAO index is from the NOAA Climate Prediction Center (CPC).

[Figure]

**Figure R3. Correlation coefficients between NAO index and zonal wind variations in April over the period 1984–2016 at 200 hPa (a), 500 hPa (c), and 850 hPa (e). Dots denote significance at the 95% confidence level, according to Student's _t_-test. The blue square indicates the area shown in Fig. R2. Before performing the analysis, the seasonal cycle and long-term linear trend were removed from the original datasets. (b, d, f) As for (a, c, e), but for geopotential height. The NAO index is from the NOAA Climate Prediction Center (CPC), and wind and geopotential height are from NCEP2.**

[Figure]

**Figure R4. Differences in composite April winds (vectors, m/s, from NCEP2) between positive and negative NAO anomaly events at 200 hPa (a), 500 hPa (b), and 850 hPa (c) for 1984–2016. Colored regions are statistically significant at the 90% (light yellow) and 95% (dark yellow) confidence levels. The seasonal cycle and long-term linear trend were removed from the original dataset. The NAO anomaly events are selected based on Table R1 below.**

**Table R1. Positive (left column) and negative (right column) NAO anomaly events in April for the period 1984–2016. Positive and negative April NAO anomaly events are defined using a normalized time series of April NAO variations from 1984 to 2016. Values larger than 1 standard deviation are defined as positive NAO anomaly events, and those below –1 standard deviation are defined as negative NAO anomaly events.**

| Positive NAO anomaly events | Negative NAO anomaly events |
|---|---|
| 1987, 1990, 1992, 2011 | 1988, 1995, 1997, 1998, 1999, 2008, 2010 |

**References:**

Calvo, N., Polvani, L. M., and Solomon, S.: On the surface impact of Arctic stratospheric ozone extremes. Environ. Res. Lett., 10, 094003, 2015.

Baldwin, M. P. and Dunkerton, T. J.: Stratospheric harbingers of anomalous weather regimes. Science, 294, 581–584, doi:10.1126/science.1063315, 2001.

Black, R. X., Mcdaniel. B. A., Robinson, W. A.: Stratosphere Troposphere Coupling during Spring Onset. J. Climate, 19, 4891-4901, 2005.

Black, R. X. and Mcdaniel, B. A.: SubMonthly polar vortex variability and stratosphere-troposphere coupling in the Arctic. J. Climate, 22, 5886-5901, 2009.

Black, R. X., Mcdaniel, B. A.: The Dynamics of Northern Hemisphere Stratospheric Final Warming Events. Journal of the Atmospheric Sciences, 64, 2932-2946, 2006.

Ivy, D. J., Solomon, S., Calvo, N., and Thompson, D. W.: Observed connections of Arctic stratospheric ozone extremes to Northern Hemisphere surface climate. Environ. Res. Lett., 12, 024004, 2017.

Smith, K. L. and Polvani, L. M.: The surface impacts of Arctic stratospheric ozone anomalies. Environ. Res. Lett., 9, 074015, 2014.

Xie, F., Li, J., Tian, W., Fu, Q., Jin, F.-F., Hu, Y., Zhang, J., Wang, W., Sun, C., Feng, J., Yang, Y., and Ding, R.: A connection from Arctic stratospheric ozone to El Niño-Southern Oscillation. Environ. Res. Lett., 11, 124026, 2016.

Xie F., Li, J., Zhang, J., Tian, W., Hu, Y., Zhao, S., Sun, C., Ding, R., Feng, J, and Yang, Y.: Variations in North Pacific Sea Surface Temperature Caused by Arctic Stratospheric Ozone Anomalies. Environ. Res. Lett., 12, 114023, 2017.

*3.2 Additionally, given that the SST anomalies shown in Fig. 10 strongly resemble the Pacific Decadal Oscillation (PDO), the same patterns should be examined for the PDO and ENSO. With such a small sample size of years used in the analysis (Table 2), the authors could simply be sampling concurrent SST variability. Given that most previous studies on this subject see the strongest anomalies in the North Atlantic sector, the fact that all of the anomalies are in the Pacific in this study makes me concerned that Pacific*

*SST variability is being aliased into the analysis.*

**Response: We thank the reviewer for this comment. Figure R5 shows the correlation coefficients between April SST variations and (Fig. R5a) April precipitation in the northwestern United States, (Fig. R5b) March ASO, (Fig. R5c) April PDO, and (Fig. R5d) April Nino 3.4 indices. Figure R5a and R5b are Fig. 10 in the initial manuscript. Comparing Fig. R5a with Fig. R5b–d, the pattern of correlation coefficients in Fig. R5a is closer to the pattern shown in Fig. R5b (ASO and SST).**

[Figure]

**Figure R5. Correlation coefficients between April SST variations and (a) April precipitation in the northwestern United States, (b) March ASO, (c) April PDO, (d) and April Nino 3.4 indices for 1984–2016. Before performing the analysis, the seasonal cycle and linear trend were removed from the original datasets.**

**To provide a more quantitative answer to the question, Table R2 lists the spatial correlation coefficients between Fig. R5a and the patterns in Fig. R5b–d. The highest spatial correlation coefficient is obtained for the patterns in Fig. R5a and Fig. R5b. Table R3 further lists the correlation coefficients between the time series of April precipitation in the northwestern United States and March ASO, April PDO, and April Nino 3.4 indices. The highest correlation coefficient is between precipitation and ASO. The above results indicate that the SST anomalies shown in Fig. 10a in the initial manuscript are more likely related to ASO.**

The mechanism by which March ASO variations affect April SST in the North Pacific was studied in detail by Xie et al. (2017) using observational data and model simulations. For a full response to this question, please see point #6 below.

Table R2. Spatial correlation coefficients for the patterns over the North Pacific only (124.5°E–100.5°W, 20.5°N–65.5°N) in Fig. R5a–d.

| Patterns | −ASO (Fig. R5b) | PDO (Fig. R5c) | ENSO (Fig. R5d) |
|---|---|---|---|
| Precipitation (Fig. R5a) | 0.72 | 0.40 | 0.31 |

Table R3. Correlation coefficients between time series of April precipitation in the northwestern United States, and March ASO, April PDO, and April Nino 3.4 indices for 1984–2016.

| Time series | −ASO | PDO | ENSO |
|---|---|---|---|
| Precipitation | 0.55 | 0.25 | 0.20 |

Reference:

Xie F., Li, J., Zhang, J., Tian, W., Hu, Y., Zhao, S., Sun, C., Ding, R., Feng, J, and Yang, Y.: Variations in North Pacific Sea Surface Temperature Caused by Arctic Stratospheric Ozone Anomalies. Environ. Res. Lett., 12, 114023, 2017.

*4. As a related point, variations in March ASO should be linked closely to the timing of the seasonal breakdown of the NH stratospheric polar vortex. Black has examined this issue in detail in a series of papers (e.g., Black et al. 2006). Again, the authors need to better contextualize their results in the context of the past literature, which emphasizes the North Atlantic.*

**Response: Thanks very much for the comment. After reading the literatures**

provided by reviewer, we realized that there are indeed many omissions in this manuscript. In the revised version, we made up for these missing knowledge points. Those some important references are also cited.

*5. A consistent measure of statistical significance needs to be provided throughout the paper. Some figures show a 90% level, others a 95% level, and some show no significance at all (model results). 90% is a fairly weak threshold for statistical significance (1 in 10 chance that the point is significant by chance). I would recommend using the 95% level, or at least showing both the 90% and 95% levels (as is shown in Fig. 6).*

**Response: Thank you for this comment. In the revised paper, we used 95% confidence level throughout the paper and added significance test to the model results.**

*6.1 Related to point #3 above, how can we be sure that the SST anomalies in Fig. 10 are in fact caused by the stratospheric anomalies? They seem inconsistent with the wind anomalies in Fig. 4 (enhanced air-sea fluxes and cooling should occur in regions of enhanced westerlies). Some physical mechanism linking ASO to the SST anomalies needs to be provided.*

**Response: Thanks very much for the important comment. Xie et al. (2017) have recently explained why the ASO has a lagged impact on the sea surface temperature in the North Pacific mid–high latitudes (Fig. R6). They found that the stratospheric circulation anomalies caused by ASO changes can rapidly extend to the lower troposphere in the Northern Hemisphere high latitudes; however, the lower troposphere high-latitude circulation anomalies take about 1 month to propagate to the North Pacific mid-latitudes. The key findings of Xie et al. (2017) are as follows:**

**Xie et al. (2017) used composite analysis and wave ray theory to understand the lagged process. Figure R7 shows the composite changes in circulation on a daily time scale during ASO decrease and increase events (this is Figure 3 in Xie**

et al. 2017). Figure R7a and b indicate that the composite Arctic stratospheric circulation anomalies during ASO anomaly events propagate downward to the high latitudes of the lower troposphere in a few days. The anomalies reaching the troposphere continue to propagate meridionally toward the northern lower and middle latitudes along the 180° to 120°W longitude zone (Fig. R7c and d). This southward propagation takes about 1 month. This phenomenon can be seen in both the ASO decrease and increase events (Fig. R7a/c and b/d).

To study in more detail the horizontal propagation of circulation anomalies, the ray paths of waves at 850 hPa generated by the perturbed circulation over the region 60°–90°N and 180°–120°W in March are shown in Fig. R8 (Figure 4 in Xie et al. (2017), who found that the circulation anomalies over the region 60°–90°N and 180°–120°W have the strongest simultaneous correlation with the ASO changes). The wavenumbers along these rays are between 1 and 3. The wave ray paths represent the climate teleconnections; i.e., the propagation of stationary waves in realistic flows. The calculation of the wave ray paths and application of the barotropic model are described in detail by Li et al. (2015) and Zhao et al. (2015). Xie et al. (2017) found that the Rossby waves generated by the perturbed circulation over the north polar lower troposphere in March mainly propagate southward to the central North Pacific after about 1 month (they propagate to the northern North Pacific in about 15 days). The wave ray paths are in good agreement with the composite analysis in Fig. R7.

Figures R6-8 imply that ASO changes take at least 1 month to influence North Pacific circulation and SST. For more details, see Xie et al. (2017).

Figure R9 shows the April 850 hPa zonal wind anomalies during negative ASO anomalies events and the corresponding climatology. It is found that the westerly in the middle North Pacific is significantly enhanced during negative ASO anomalies events. It agreed with Fig. R10 that a negative SST anomaly is found in the middle North Pacific. In addition, the SST anomalies forced by ASO also can be explained by NPO anomalies. As mentioned above, the variations in ASO relate to NPO anomalies. Alexander et al. (2010) and Yu and Kim (2011) reported that

anomalous surface wind associated with the NPO can force a tripole-like pattern of the surface heat flux anomalies in the North Pacific, which in turn induces a tripole SSTA pattern there (including a dipole SSTA pattern of the Victoria Mode (VM) in the North Pacific poleward of 20°N).

The further proof from full-couple climate-ocean model please sees the next Response.

[Figure]

(Figure 6c in Xie et al. 2017)

**Figure R6. Correlation coefficients in March for 1979–2015 between −ASO and SST a month later. Only regions above the 95% confidence level are colored. The ASO data are from MERRA2, SST from HadSST.**

[Figure]

(Figure 3 in Xie et al. 2017)

**Figure R7. Time–height cross-section of composite daily variations in zonal wind (averaged over 60°–90°N, 180°–120°W) and latitude–time cross-section of composite daily variations in**

zonal wind at 850 hPa (averaged over 180°–120°W) during ASO decrease events (a) and (c) and increase events (b) and (d) in March from 1979 to 2015. Winds are from NCEP2. The pink and green arrows indicate the propagation pathways of circulation anomalies.

[Figure]

(Figure 4 in Xie et al. 2017)

**Figure R8. Ray paths (green lines) at 850 hPa in March after the circulation was perturbed for 15 days (a) and 30 days (b). Red dots denote wave sources in the region 60°–90°N, 180°– 120°W. The wavenumbers along these rays are in the range 1–3. Color shading indicates the climatological flow.**

[Figure]

**Figure R9. (a) April zonal wind anomalies during negative ASO anomalies events and (b) the climatology of the zonal wind in April at 850 hPa. The negative ASO anomalies events is based on Table 2 in the manuscript. Zonal wind is from NCEP2.**

[Figure]

**Figure R10. Composite SST anomalies during negative ASO anomaly events.**

**References:**

**Alexander, M. A., Vimont, D. J., Chang, P., and Scott, J. D.: The impact of extratropical atmospheric variability on ENSO: Testing the seasonal footprinting mechanism using coupled model experiments. J. Clim., 23, 2885–2901, 2010.**

**Li Y. J., Li J., Jin F-F, and Zhao S.: Interhemispheric propagation of stationary rossby waves in a horizontally no uniform background flow. J. Atmos. Sci. 72, 3233–3256, 2015.**

**Xie F., Li, J., Zhang, J., Tian, W., Hu, Y., Zhao, S., Sun, C., Ding, R., Feng, J, and Yang, Y.: Variations in North Pacific Sea Surface Temperature Caused by Arctic Stratospheric Ozone Anomalies. Environ. Res. Lett., 12, 114023, 2017.**

**Yu, J.-Y., and Kim, S. T.: Relationships between extratropical sea level pressure variations and the central Pacific and eastern Pacific types of ENSO. J. Clim., 24, 708–720, 2011.**

**Zhao S., Li J., and Li Y. J.: Dynamics of an interhemispheric teleconnection across the critical latitude through a southerly duct during boreal winter. J. Clim. 28, 7437–7456, 2015.**

*6.2 Without prescribing ASO anomalies in a fully coupled model (with interactive SSTs), it's difficult to conclusively establish that the SST anomalies can in fact be forced by ASO.*

**Response: To further confirm the leading effect of ASO on North Pacific SST, Xie et al. (2017) used the National Center for Atmospheric Research's Community Earth System Model (CESM) version 1.0.6 to simulate this process, which is a fully**

coupled global climate model that incorporates an interactive atmosphere (CAM/WACCM) component, ocean (POP2), land (CLM4), and sea ice (CICE). For the atmospheric component, they used the Whole Atmosphere Community Climate Model (WACCM), version 4 (Marsh et al 2013), which has a finite volume dynamical core and it extends from the surface to approximately 140 km. For their study, they disabled the interactive chemistry.

The transient experiment (E1) performed by CESM with the fully coupled ocean incorporating both natural and anthropogenic external forcings, including spectrally resolved solar variability (Lean et al 2005), transient greenhouse gases (GHGs) (from scenario A1B of IPCC 2001), volcanic aerosols (from the Stratospheric Processes and their Role in Climate (SPARC) Chemistry–Climate Model Validation (CCMVal) REF-B2 scenario recommendations), a nudged quasi-biennial oscillation (QBO) (the time series in CESM is determined from the observed climatology over the period 1955–2005), and specified ozone forcing derived from the CMIP5 ensemble mean ozone output. E1 is a historical simulation covering the period 1955–2005. All the forcing data used in their study are available from the CESM model input data repository.

The experiment E1, covering the period 1955–2005 and with the specified ASO forcing applied to the CESM, captures the leading effect of the specified ASO anomalies on the North Pacific (Fig. R11). The VM-like pattern SST anomalies that appear over the North Pacific in April. This simulated result is similar to the observations (Figs. R6 and R10). Note that the ozone forcing is specified in the simulation and SST is output; therefore, the relationship between ASO and SST variations could only be caused by North Pacific SST anomalies related to the ASO changes.

More descriptions of the lagged impact of ASO on North Pacific SST anomalies shown in Xie et al. (2017) are added in the revised paper. Please see lines 284–290.

[Figure]

(Figure 7d in Xie et al. 2017)

**Figure R11. Correlation coefficients between the specified –ASO in March and SST in April for the period 1955–2005 in the model simulation experiment. Only regions above the 95% confidence level are colored. All quantities were detrended before correlation.**

*Minor Revisions:*

*7. The following sentence structure used in the abstract (and elsewhere in the paper) is very difficult to read: "An increase (decrease) .... results in enhanced (weakened) ... but weakened (enhanced) ...   facilitating (impeding) ..." Please consider eliminating the words in parentheses, or using a difficult format to convey this information. It's confusing to discuss both polarities (both an increase and decrease in ozone) within the same sentence structure.*

**Response: Thanks for the comment. We have used a new format to convey this information in the whole manuscript.**

*8. Line 37: The circulation changes mostly occurred in the late 20th century, not the early 21st century, as the ozone hole was increasing in size from the 1980s until around the year 2000. Since that time, the ozone hole has stabilized in size, and may in fact be starting to recover (see Solomon et al. 2017).*

**Response: Revised. Thanks.**

*9. Lines 44–45. See Fig. 3 in Kang et al. (2011). The precipitation changes associated with Antarctic stratospheric ozone depletion are more accurately described as an increase in the subtropics and high latitudes, and a decrease at mid-latitudes.*

**Response: Revised. Thank you.**

*10. Lines 55–57: This explanation of the surface temperature anomalies associated with Antarctic ozone depletion is not consistent with previous literature. See discussion in Thompson et al. (2011) and the references therein. The surface temperature anomalies are linked to how the poleward circulation shift associated with the ozone hole affects localized wind patterns (and associated thermal advection) at each location.*

**Response: We thank the reviewer for this comment. We have modified this section as follows (lines 38-41 in the revised manuscript):**

"***The poleward circulation shift would cause surface temperature anomalies by affecting localized wind patterns and associated thermal advection (Son et al., 2010; Thompson et al. 2011; Feldstein, 2011).***"

**References:**

**Thompson, D. W. J., Solomon, S., Kushner, P.'J., England, M. H., Grise, K. M., and Karoly, D. J.: Signatures of the Antarctic ozone hole in Southern Hemisphere surface climate change, Nature Geosci., 4, 741–749, doi:10.1038/NGEO1296, 2011.**

*11. Line 110: The vertical pressure range (100–50 hPa) contradicts that in footnote #2 of Table 1 (150–50 hPa). Please correct.*

**Response: Corrected. Thanks.**

*12. Line 142: If SSTs are specified, the term "coupled" here is misleading. Follow convention in the literature, I would recommend using the term "coupled" only if the atmosphere model is fully coupled to an interactive ocean model.*

**Response: Removed the term "coupled". Thanks.**

*13. Line 148: This statement seems to contradict the statement on line 143. The model has middle atmospheric chemistry, yet the model does not include interactive chemistry. This needs to be clarified.*

**Response: Thanks for the comment. There are two schemes to run WACCM4, one is WACCM4-MOZART (including interactive chemistry), and one is WACCM4-GHG (disable interactive chemistry). Our study used the latter scheme. We are sorry that it is not clear here. It has been revised in the revised manuscript.**

*14. Lines 151, 154: The text refers to a reference period of 1980–2015, while Table 1 refers to a reference period of 1995–2005. This needs to be clarified and standardized throughout the paper.*

**Response: Revised. Thanks.**

*15. Line 164: I would use "break down" rather than "rupture" here to be consistent with terminology in previous literature.*

**Response: Revised. Thanks.**

*16. Line 167: This lead time is not unique to NH stratospheric ozone perturbations. It is consistent with the tropospheric anomalies associated with NH sudden stratospheric warmings (Baldwin and Dunkerton 2001) and SH stratospheric ozone depletion (Thompson and Solomon 2002).*

**Response: Thanks for this comment. Here, we added the some content in the revised paper. See lines 165-169 in the revised manuscript.**

*"These studies pointed out that the changes in ASO affect the tropospheric climate with a lead of about 1–2 months, which is similar to the troposphere response to the Northern Hemisphere sudden stratospheric warmings (Baldwin and Dunkerton 2001; Black et al., 2005, 2006, 2009) and Southern Hemisphere stratospheric ozone depletion (Thompson and Solomon 2002)."*

**References:**

**Baldwin, M. P. and Dunkerton, T. J.: Stratospheric harbingers of anomalous weather regimes, Science, 294, 581–584, doi:10.1126/science.1063315, 2001.**

Black, R. X., Mcdaniel. B. A., Robinson, W. A.: Stratosphere Troposphere Coupling during Spring Onset. J. Climate, 19, 4891-4901, 2005.

Black, R. X. and Mcdaniel, B. A.: SubMonthly polar vortex variability and stratosphere-troposphere coupling in the Arctic. J. Climate, 22, 5886-5901, 2009.

Black, R. X., Mcdaniel, B. A.: The Dynamics of Northern Hemisphere Stratospheric Final Warming Events. Journal of the Atmospheric Sciences, 64, 2932-2946, 2006.

Thompson, D. W. J. and Solomon, S.: Interpretation of recent Southern Hemisphere climate change, Science, 296, 895–899, doi:10.1126/science.1069270, 2002.

*17. Lines 236–240: This statement is not consistent with the figures. Figures 4–6 show a barotropic circulation response (same sign throughout the depth of the troposphere), with an anomalous cyclone over western North America at all levels (Fig. 6).*

*18. Lines 248–250: How so? I don't understand the dynamical basis for this statement.*

**Response: We thank the reviewer for the two comments. We apologize for the incorrect statement here. In the revised manuscript, we have rewritten the paragraph in lines 231–250 as follows (see lines 199–232 in the revised manuscript):**

*"Figure 3 shows the correlation coefficients between March ASO anomalies and April zonal wind variations at 200, 500, and 850 hPa, respectively. The spatial distribution of significant correlation coefficients over the North Pacific exhibits a tripolar mode with a zonal distribution at 200 and 500 hPa; i.e. a positive correlation in the high and low latitudes in the North Pacific and a negative correlation in mid-latitudes. This implies that the increase in ASO can result in enhanced westerlies in the high and low latitudes of the North Pacific but weakened westerlies in the mid-latitudes, corresponding to the weakened Aleutian Low in April, and vice versa for the decrease in ASO. The Aleutian Low acts as a bridge connecting variations in ASO and circulation anomalies over the North Pacific (Xie et al., 2017a). At 850 hPa, the anomalous circulation signal in the low latitudes of the North Pacific has weakened*

*and disappeared. It is evident that the anomalous changes in the zonal wind over the North Pacific can extend westward to East Asia. Xie et al. (2018) identified the effect of spring ASO changes on spring precipitation in China. Note that the weakened westerlies in the mid-latitudes and the enhanced westerlies at low latitudes can also extend eastward to the western United States. This kind of circulation anomaly corresponds to two barotropic structures; i.e., an anomalous anticyclone in the Northeast Pacific and a cyclone in the southwestern United States at 500 hPa and 200 hPa. Coincidentally, the northwestern United States is located to the north of the intersection of the anticyclone and cyclone, corresponding to convergence of the airflow at high levels, which may lead to downwelling in the northwestern United States, and vice versa for negative March ASO anomalies.*

*To further validate our inference regarding the response of the circulation in the western United States to ASO changes, we analyze the differences between April horizontal wind anomalies during positive and negative March ASO anomaly events at 200, 500, and 850 hPa (Fig. 4). As in the increased ASO case, the difference shows an anomalous anticyclone in the Northeast Pacific and an anomalous cyclone in the southwestern United States. This kind of circulation anomaly over the southwestern United States enhances cold and dry airflow from the North American continent to the North Pacific, reducing the water vapor concentration in the air over the western United States and possibly reducing April precipitation in the northwestern United States. In addition, the northwestern United States is located to the north of the intersection of the anticyclone and cyclone, suggesting downwelling flow in the region."*

*19. Lines 254, 258, 365: Reanalyses cannot adequately resolve convective activity. The*

*anomalous downwelling here is associated with synoptic-scale processes (see positive geopotential heights in northeast Pacific in Fig. 5). The pattern in Fig. 7a should closely correspond to sea-level pressure anomalies (a surface high in the northwestern United States and a surface low in the southwestern United States).*

**Response: We thank the reviewer for this comment. The term "convective activity" used here may not be appropriate. We have used "downwelling" instead of "convective activity" in the revised paper.**

**Figure R12 shows the differences in composite April sea-level pressure anomalies between positive and negative ASO anomaly events for 1984–2016. The result agrees with the reviewer's speculation; i.e., a surface high in the northwestern United States and a surface low in the southwestern United States.**

[Figure]

**Figure R12. Differences in composite April sea-level pressure anomalies between positive and negative ASO anomaly events for 1984–2016. Before performing the analysis, the seasonal cycle and linear trend were removed from the original dataset. The ASO anomaly events are selected from Table 2 in the manuscript. The SLP (Pa) dataset is from the UK Met Office Hadley Centre.**

*20. Line 282: It doesn't look like opposite to me … just shifted a little further to the north in the model than in the observations (which, of course, would make a difference for regional impacts as the authors nicely state on the subsequent lines).*

**Response: Revised. Thanks.**

*21. Line 301: The SST pattern looks a lot like the Pacific Decadal Oscillation (PDO) or the North Pacific Mode (Hartmann 2015). How well correlated is the time series of the "Victoria Mode" with these modes?*

**Response: Thanks for the comment. Please see the Table R4.**

**Table R4. Correlation coefficients among the time series of April VM, April PDO, and April Nino 3.4 indices for 1984–2016.**

| Time series | PDO | ENSO |
|:---:|:---:|:---:|
| VM | -0.18 | 0.09 |

*22. All figures: I think it's unnecessary to show both the correlations and composite differences (Figs. 1 and 3, left and right columns of Fig. 4-5), as they basically convey identical information.*

**Response: Thanks for this comment. Only the correlation results remain in the revised manuscript.**

*23. Figures 1 and 3: It's difficult to interpret these patterns with so much of the map left blank. I would recommend showing the correlation coefficients for the entire map, and stippling those regions that are statistically significant.*

**Response: Modified, thanks. See following Fig. R13 (this is Figure 1 in the manuscript). Figure 3 in the revised manuscript has been deleted.**

[Figure]

**Figure R13. Correlation coefficients between March ASO and April precipitation variations**

calculated from SWOOSH (a, b) and GOZCARDS (c, d) ozone, and GPCC (a, c) and GPCP (b, d) rainfall for the period 1984–2016. Dots denote significance at the 95% confidence level, according to Student's t-test. The long-term linear trend and seasonal cycle in all variables were removed before the correlation analysis.

*24. Figure 2: What are the dashed black lines? A measure of statistical significance?*

**Response: This point is right, thanks. The dashed blacked lines refer to the correlation coefficient that is significance at 95% confidence level. In the revised paper, we added it in the caption of Fig. 2.**

*25. Figures 4–5: Is it necessary to show both geopotential heights and zonal wind? Both figures convey exactly the same information (via geostrophic balance).*

**Response: Deleted Figure 5. Thanks.**

*26. Figure 7: It would be good to clarify that blue is upward motion and red is downward motion.*

**Response: Clarified it in the caption of Figure 7. Thanks.**

*27. Figure 12: Because the model has prescribed SSTs, how do you know the model SSTs associated with ASO anomalies? Are these some version of the observed SST anomalies as they don't look exactly like those in Fig. 10?*

**Response: We thank the reviewer for this comment. The SST anomalies used to force the model (Fig. 12; that is Fig. 9 in the revised manuscript) are composite SST anomalies for negative and positive ASO anomaly events. The mechanism by which ASO influences the North Pacific SST is discussed in the responses to points 6.1 and 6.2. The pattern in Fig. 12 (Fig. 9 in the revised manuscript) is not exactly the same as that in Fig. 10 (Fig. 8 in the revised manuscript), which may reflect the fact that the results in Fig. 12 (Fig. 9 in the revised manuscript) were obtained from composite analysis whereas those in Fig. 10 (Fig. 8 in the revised manuscript) were from correlation analysis.**

*Typos:*

1. *Line 96: central of China -> central China*

2. *Line 134: regarding -> regarded*

3. *Line 147: is at -> are at*

4. *Lines 173, 354: Washington and Oregon states -> Washington and Oregon*

5. *Line 176: the Fig. 1 -> Fig. 1*

6. *Line 259: enhances -> weakens*

7. *Table 1, R4–R7: a SST anomalies -> SST anomalies*

**Response: All revised. Thanks.**

---

## Author Comment (AC2) · 13 Sep 2018

**Responses to Referees' Comments**

**Effects of Arctic stratospheric ozone changes on spring precipitation in the northwestern United States (ACP-2018-414)**

Xuan Ma, Fei Xie, Jianping Li, Xinlong Zheng, Wenshou Tian,
Ruiqiang Ding, Cheng Sun, and Jiankai Zhang

September 2018

**Response to Referee 2**

*The manuscript presents a well-designed study of the effects of variability in springtime Arctic stratospheric ozone (ASO) on the tropospheric circulation over the Pacific basin, extending into the north-west United States. The authors present statistical relationships between a variety of physical climate variables and ASO in observations, finding an inverse correlation between ASO anomalies and March precipitation over the north-west United States, and then explore the causality with a number of WACCM model simulations using anomalies applied to the prescribed ozone and sea-surface temperatures used in the model. The model simulations provide convincing evidence that the combined effect of the ASO anomalies and correlated changes in sea-surface temperatures over the Pacific can reproduce the observed pattern of changes in winds and precipitation. The study is well thought out and presented and I have no serious concerns about the methodology.*

*The one significant missing aspect to the manuscript is the way the authors discuss ozone variability and the effects of ozone variability on dynamics as a completely independent forcing. The model simulations are convincing in that the specified ozone can be modified and the impact on the dynamics can be estimated in a one-way cause-and-effect manner. But in the real atmosphere there is a very tight coupling between dynamical modes of variability and Arctic ozone. Variability in the amount of planetary wave forcing from the troposphere has a direct connection to the strength of the Brewer-Dobson circulation and the amount of poleward ozone transport each year. And the occurrence of Sudden Stratospheric Warmings in the late winter or early spring can determine whether polar stratospheric temperatures cold enough for heterogenous chemistry on polar stratospheric clouds will occur and produce significant chemical ozone destruction in the Arctic. I think there are two important implications for the manuscript under consideration here. One is that the observation-based analysis must discuss the strong coupling between dynamical variability and ozone variability and must recognize that the correlations of certain physical variables with ozone also*

*reflect correlations with other aspects of dynamical variability. And second, I believe the authors cannot state that the Victoria Mode anomalies in Pacific sea-surface temperatures are caused by, as opposed to being associated with, the ASO anomalies.*

**Response: We thank the reviewer for taking the time to assess the manuscript and we sincerely appreciate the reviewer's helpful comments, which have greatly improved the paper. We have revised the manuscript carefully according to the reviewer's comments and suggestions.**

**We agree with the reviewer's opinion that the spring ASO variations are related to changes in the winter Arctic stratospheric vortex (SPV). The strength of the SPV can affect ASO, and then ASO affects tropospheric teleconnection and precipitation in the northwestern United States (indirect effect of SPV). The strength of the SPV may also have a direct leading effect on tropospheric teleconnection and precipitation in the northwestern United States. There is a tight coupling between dynamical modes of variability and ASO. In this study, we have not thought of a better way to separate the two effects on precipitation. Thus, the thrust of this study is to at least recognize that the ASO changes may affect precipitation in the northwestern United States. From the analysis of observational data, we find that the ASO has a leading relationship with spring precipitation in the northwestern United States. In addition, this relationship can be reproduced in simulations by abnormal ASO forcing. This implies that the variations in spring ASO can force the observed tropospheric circulation and precipitation anomalies in the northwestern United States.**

**Figure RR1 shows the correlation coefficients between the February SPV (multiplied by –1) index and April 200 hPa zonal wind and precipitation variations (Fig. RR1a and b), and between March ASO and April 200 hPa zonal wind and precipitation (Fig. RR1c and d). The SPV index is defined as the strength of the stratospheric polar vortex, following Zhang et al. (2018). Although the patterns of correlation coefficients in Fig. RR1 are similar, the ASO variations are much closer than the strength of the stratospheric polar vortex to the variations in 200 hPa zonal wind and precipitation. Fig. RR1 indicates indirect and direct effects of winter SPV on spring tropospheric climate. Since the coupling between dynamical**

and radiative processes in spring is strong, the connection between winter SPV and spring tropospheric circulation seems weaker than that between the spring ASO and tropospheric circulation.

[Figure]

Figure RR1. (a) Correlation coefficients between the February –SPV ($10^5$ K m$^2$ kg$^{-1}$ s$^{-1}$) index defined by Zhang et al. (2018) and April zonal wind variations at 200 hPa for 1984–2016. (b) Correlation coefficients between February –SPV index and April precipitation variations. (c) and (d) As for (a) and (b), but between March ASO and April 200 hPa zonal wind and April precipitation variations. Dots denote significance at the 95% confidence level, according to Student's *t*-test. The long-term linear trend and seasonal cycle in all variables were removed before the correlation analysis. The ASO data is from SWOOSH, zonal wind from NCEP2, and precipitation from GPCP.

We apologize for the lack of clarity that led the reviewer to feel that the manuscript overemphasized the influence of ozone on stratosphere and troposphere coupling in spring. In this study, we want to state that the ASO changes possibly influence precipitation in the northwestern United States, emphasizing the influence of stratospheric ozone on tropospheric regional climate. The direct and indirect impacts of SPV on precipitation in the northwestern United States and the effect of the strong coupling between dynamical and ozone

variability are indeed important issues that we will examine in future work.

We have made this point clearer in the revised manuscript. The Fig. RR1 and relevant discussion have been added to the discussion section in the revised manuscript. See lines 368–388.

Xie et al. (2017) explained why the ASO has a lagged impact on the circulation and sea surface temperature in the North Pacific mid–high latitudes based on observations and a fully coupled climate–ocean model. Detailed responses are given below.

References:

Xie F., Li, J., Zhang, J., Tian, W., Hu, Y., Zhao, S., Sun, C., Ding, R., Feng, J, and Yang, Y.: Variations in North Pacific Sea Surface Temperature Caused by Arctic Stratospheric Ozone Anomalies. Environ. Res. Lett., 12, 114023, 2017.

Zhang J., et al.: Stratospheric ozone loss over the Eurasian continent induced by the polar vortex shift, Nat. Commun., 9, 206, 2018.

*As given below in the minor comments, in a few places through the manuscript the differences in the circulation between different WACCM experiments are described in very direct ways. It would be much more illustrative for the reader if these changes could be associated with changes in the position of significant climatological features, in a similar way that the Antarctic wind changes can be summarized as a pole-ward shift of the jet.*

**Response: Thank you for the good suggestion. We also recognize this problem. We have described those features in the Figures in a more physical and professional language. Please see the manuscript in detail.**

*Minor Comments:*

*1. Lines 15 – 18: Following my concerns about correlation and causality, the sentence 'In addition, the ASO changes cause sea surface temperature anomalies over the North Pacific that would cooperate with the ASO changes to modify the circulation anomalies*

*over the northwestern US.' should be softened.*

**Response: Thanks to the comment. This sentence has been modified as follows:**

***"In addition, sea surface temperature anomalies over the North Pacific, which may be related to the ASO changes, would cooperate with the ASO changes to modify the circulation anomalies over the northwestern United States."***

**Xie et al. (2017) have recently explained why the ASO has a lagged impact on the sea surface temperature in the North Pacific mid–high latitudes (Fig. RR2). They found that the stratospheric circulation anomalies caused by ASO changes can rapidly extend to the lower troposphere in the Northern Hemisphere high latitudes; however, the lower troposphere high-latitude circulation anomalies take about 1 month to propagate to the North Pacific mid-latitudes. The key findings of Xie et al. (2017) are as follows:**

**Xie et al. (2017) used composite analysis and wave ray theory to understand the lagged process. Figure RR3 shows the composite changes in circulation on a daily time scale during ASO decrease and increase events (this is Figure 3 in Xie et al. 2017). Figure RR3a and b indicate that the composite Arctic stratospheric circulation anomalies during ASO anomaly events propagate downward to the high latitudes of the lower troposphere in a few days. The anomalies reaching the troposphere continue to propagate meridionally toward the northern lower and middle latitudes along the 180° to 120°W longitude zone (Fig. RR3c and d). This southward propagation takes about 1 month. This phenomenon can be seen in both the ASO decrease and increase events (Fig. RR3a/c and b/d).**

**To study in more detail the horizontal propagation of circulation anomalies, the ray paths of waves at 850 hPa generated by the perturbed circulation over the region 60°–90°N and 180°–120°W in March are shown in Fig. RR4 (Figure 4 in Xie et al. (2017), who found that the circulation anomalies over the region 60°–90°N and 180°–120°W have the strongest simultaneous correlation with the ASO changes). The wavenumbers along these rays are between 1 and 3. The wave ray**

paths represent the climate teleconnections; i.e., the propagation of stationary waves in realistic flows. The calculation of the wave ray paths and application of the barotropic model are described in detail by Li et al. (2015) and Zhao et al. (2015). Xie et al. (2017) found that the Rossby waves generated by the perturbed circulation over the north polar lower troposphere in March mainly propagate southward to the central North Pacific after about 1 month (they propagate to the northern North Pacific in about 15 days). The wave ray paths are in good agreement with the composite analysis in Fig. RR3.

Figures RR2-4 imply that ASO changes take at least 1 month to influence North Pacific circulation and SST.

[Figure]

**(Figure 6c in Xie et al. 2017)**

Figure RR2. Correlation coefficients in March for 1979–2015 between –ASO and SST a month later. Only regions above the 95% confidence level are colored. The ASO data are from MERRA2, SST from HadSST.

[Figure]

(Figure 3 in Xie et al. 2017)

**Figure RR3. Time–height cross-section of composite daily variations in zonal wind (averaged over 60°–90°N, 180°–120°W) and latitude–time cross-section of composite daily variations in zonal wind at 850 hPa (averaged over 180°–120°W) during ASO decrease events (a) and (c) and increase events (b) and (d) in March from 1979 to 2015. Winds are from NCEP2. The pink and green arrows indicate the propagation pathways of circulation anomalies.**

[Figure]

(Figure 4 in Xie et al. 2017)

**Figure RR4. Ray paths (green lines) at 850 hPa in March after the circulation was perturbed for 15 days (a) and 30 days (b). Red dots denote wave sources in the region 60°–90°N, 180°–120°W. The wavenumbers along these rays are in the range 1–3. Color shading indicates the climatological flow.**

The further proof from full-couple climate-ocean model, Xie et al. (2017) used

the National Center for Atmospheric Research's Community Earth System Model (CESM) version 1.0.6 to simulate this process, which is a fully coupled global climate model that incorporates an interactive atmosphere (CAM/WACCM) component, ocean (POP2), land (CLM4), and sea ice (CICE). For the atmospheric component, they used the Whole Atmosphere Community Climate Model (WACCM), version 4 (Marsh et al. 2013), which has a finite volume dynamical core and it extends from the surface to approximately 140 km. For their study, they disabled the interactive chemistry.

The transient experiment (E1) performed by CESM with the fully coupled ocean incorporating both natural and anthropogenic external forcings, including spectrally resolved solar variability (Lean et al. 2005), transient greenhouse gases (GHGs) (from scenario A1B of IPCC 2001), volcanic aerosols (from the Stratospheric Processes and their Role in Climate (SPARC) Chemistry–Climate Model Validation (CCMVal) REF-B2 scenario recommendations), a nudged quasi-biennial oscillation (QBO) (the time series in CESM is determined from the observed climatology over the period 1955–2005), and specified ozone forcing derived from the CMIP5 ensemble mean ozone output. E1 is a historical simulation covering the period 1955–2005. All the forcing data used in their study are available from the CESM model input data repository.

The experiment E1, covering the period 1955–2005 and with the specified ASO forcing applied to the CESM, captures the leading effect of the specified ASO anomalies on the North Pacific (Fig. RR5). The VM-like pattern SST anomalies that appear over the North Pacific in April. This simulated result is similar to the observations (Figs. RR2). Note that the ozone forcing is specified in the simulation and SST is output; therefore, the relationship between ASO and SST variations could only be caused by North Pacific SST anomalies related to the ASO changes.

More descriptions of the lagged impact of ASO on North Pacific SST anomalies shown in Xie et al. (2017) are added in the revised paper. Please see lines 284–290.

[Figure]

(Figure 7d in Xie et al. 2017)

**Figure RR5. Correlation coefficients between the specified –ASO in March and SST in April for the period 1955–2005 in the model simulation experiment. Only regions above the 95% confidence level are colored. All quantities were detrended before correlation.**

**References:**

**Lean, J., Rottman, G., Harder, J. and Kopp, G.: SORCE contributions to new understanding of global change and solar variability. Sol. Phys. 230 27–53, 2005.**

**Li Y. J., Li J., Jin F-F, and Zhao S.: Interhemispheric propagation of stationary rossby waves in a horizontally no uniform background flow. J. Atmos. Sci. 72, 3233–3256, 2015.**

**Marsh, D. R., Mills, M. J., Kinnison, D. E., Lamarque, J. F., Calvo, N., and Polvani, L. M.: Climate Change from 1850 to 2005 Simulated in CESM1(WACCM), J. Climate, 26, 7372–7391, doi:10.1175/JCLI-D-12-00558.1, 2013.**

**Xie F., Li, J., Zhang, J., Tian, W., Hu, Y., Zhao, S., Sun, C., Ding, R., Feng, J, and Yang, Y.: Variations in North Pacific Sea Surface Temperature Caused by Arctic Stratospheric Ozone Anomalies. Environ. Res. Lett., 12, 114023, 2017.**

**Zhao S., Li J., and Li Y. J.: Dynamics of an interhemispheric teleconnection across the critical latitude through a southerly duct during boreal winter. J. Clim. 28, 7437–7456, 2015.**

*2. Lines 109 – 111: As stated here, the ASO is calculated as an anomaly after removing the annual cycle and trend. I would imagine the long-term trend is predominately due to the rise in ozone depleting substances. Why was the trend removed from the calculation of ASO, as I would think the March ASO anomaly related to ozone depletion would be part of the signal you are looking for? And is the trend calculated as a single linear trend across the entire period or some measure that is related to halogen loading*

*in the stratosphere such as Equivalent Effective Stratospheric Chlorine (EESC)? As the period analysed is 1984 – 2015, or so, this would include both the rapid increase in EESC up to ~2000 and the plateau or slow decline since then and a single linear trend across the entire period would be a less than ideal estimate of the forced response.*

**Response: We thank the reviewer for this comment. Figure RR6 shows the standardized time series of the original March ASO index (black line), the index after removal of the linear trend across the entire period (blue line), and that after removal of the EESC signal (red line). The correlation coefficients between these ASO time series are listed in Table RR1. These three ASO time series are very similar, and the correlation coefficients are all above 0.95 and significant at the 95% confidence level.**

**To further assess the response of April circulation variations to ASO changes with and without the linear trend and EESC signal, Figure RR7 shows the correlation coefficients between these three ASO time series and April zonal wind variations. All three March ASO indices are significantly correlated with April zonal wind variations over the North Pacific, and their patterns are similar in each case. This implies that the trend of ASO from 1984 to 2016 does not affect the main conclusions of this study.**

[Figure]

**Figure RR6. ASO represented by a standardized time series of March mean ozone from SWOOSH ozone for 1984 to 2016. Black line presents the original data; blue line shows the ASO with the linear trend removed and the red line is the ASO with the EESC signal removed.**

**Table RR1. Correlation coefficients between the three ASO time series shown in Fig. RR6.**

|                              | ASO | ASO (linear trend removed) | ASO (EESC removed) |
| ---------------------------- | --- | -------------------------- | ------------------ |
| ASO                          | 1.0 | 0.97                       | 0.98               |
| ASO (linear trend removed)   | —   | 1.0                        | 0.95               |
| ASO (EESC removed)           | —   | —                          | 1.0                |

[Figure]

**Figure RR7. (a) Correlation coefficient between the original March ASO index and April zonal wind variations (m/s, from NCEP2) from 1984 to 2016 at 200 hPa. (b) and (c) As for (a), but for the ASO index with the linear trend and EESC signal removed, respectively. Dots denote significance at the 95% confidence level, according to Student's *t*-test.**

*3. Line 117: 'Another set of ozone dataset is...' sounds a bit redundant. Could I suggest 'Another set of ozone data is...'*

**Response: Revised. Thanks.**

*4. Lines 149 – 151: The statement 'The model's radiation scheme uses these conditions: fixed greenhouse gas (GHG) values, averages of emissions scenario A2 of the Intergovernmental Panel on Climate Change (IPCC) (WMO, 2003) for 1980–2015.' is*

*difficult to interpret. Is it that the fixed GHG values that were used are the 1980-2015 average from the A2 scenario? It seems a bit clearer in the text in Table 1, but there the average is said to be over 1995-2005.*

**Response: Thanks for the comment. We are sorry for the mistake. The average time is 1995-2005, which have been modified in the revised paper. Please see lines 148 – 151.**

**"The model's radiation scheme uses these conditions: fixed greenhouse gas (GHG) values (averages of emissions scenario A2 of the Intergovernmental Panel on Climate Change (WMO, 2003) over the period 1995–2005)."**

*5. Lines 212 – 214: The correlation of zonal wind anomalies with the ASO is described as: 'This implies that the increase (decrease) in ASO can result in enhanced (weakened) westerlies in the high and low latitudes of the North Pacific but weakened (enhanced) westerlies in the mid-latitudes.' The changes in southern hemisphere winds associated with ozone depletion are often described in terms of a shift of the jet that produces a dipole pattern of changes in wind. Here the authors argue that the ASO is associated with a tripole of changes in zonal wind. Do the authors have an explanation for the pattern of changes that can be related to shifts or changes in magnitude of climatological features like the Aleutian Low? And can other explanations for the changes at low latitudes, such as ENSO, be ruled out?*

**Response: We thank the reviewer for this comment. Weakened westerlies in the high-latitude North Pacific and enhanced westerlies in the mid-latitudes during negative ASO anomaly events may not imply a poleward shift of the westerlies during ASO depletion.**

**However, as discussed by Xie et al. (2017), the pattern of zonal wind anomalies associated with ASO variations is related to changes in the North Pacific Oscillation (NPO) and Aleutian Low. Figure RR8 shows the differences in composite zonal wind between positive and negative April Aleutian Low (AL) anomaly events (selected AL events refer to Table RR2). The result shows that the**

pattern of zonal wind anomalies related to the AL index is similar to that related to the ASO (see Figure 3 in the revised manuscript). This may implies that the AL acts as a bridge connecting variations in ASO and circulation anomalies over the North Pacific (This is also stated by Xie et al., 2017). In other words, the weakened westerlies in the high-latitude North Pacific and enhanced westerlies in the mid-latitudes during negative ASO anomaly events imply that the AL is enhanced when ASO is depleted, but weakened when ASO increases.

Figure RR9 is the same as Fig. RR8, but for the Nino 3.4 index. The pattern of zonal wind anomalies related to ENSO differs from that related to ASO.

The above results illustrate that the pattern of zonal wind anomalies associated with ASO variations is possibly associated with changes in the AL. The relevant content has been added to the revised manuscript (lines 204–207) as follows:

*"This implies that the increase in ASO can result in enhanced westerlies in the high and low latitudes of the North Pacific but weakened westerlies in the mid-latitudes, corresponding to the weakened Aleutian Low in April, and vice versa for the decrease in ASO."*

[Figure]

**Figure RR8. Differences in composite April zonal wind (m/s) anomalies between positive and negative AL anomaly events at 200 hPa (a), 500 hPa (b), and 850 hPa (c). Dots denote**

significance at the 95% confidence level, according to Student's *t*-test. Before performing the analysis, the seasonal cycle and linear trend were removed from the original datasets. AL anomaly events are selected using Table RR2.

[Figure]

Figure RR9. Same as Fig. RR8, but for the Nino 3.4 index.

Table RR2. Positive (left column) and negative (right column) anomaly events based on the AL and Nino 3.4 indices for the period 1984–2016.

|  | index > 1 STD | index < −1 STD |
|---|---|---|
| AL index | 1985, 1986, 1999, 2002, 2008, 2013 | 1993, 1996, 1997, 2004, 2007, 2014, 2016 |
| Nino 3.4 index | 1987, 1992, 1993, 1998, 2015, 2016 | 1985, 1989, 1999, 2000, 2008, 2011 |

References:

Xie F., Li, J., Zhang, J., Tian, W., Hu, Y., Zhao, S., Sun, C., Ding, R., Feng, J, and Yang, Y.: Variations in North Pacific Sea Surface Temperature Caused by Arctic Stratospheric Ozone Anomalies. Environ. Res. Lett., 12, 114023, 2017.

*6. Lines 236 – 240: 'This kind of circulation anomaly corresponds to an anomalous cyclone (anticyclone) in the western US in the middle and upper troposphere, which is likely associated with a strong low (high) pressure system in the middle and upper troposphere and a relatively weak high (low) pressure system in the lower troposphere.' I can see how this description fits with the pattern of wind changes shown in Figure 6, but that the pattern of changes shown in panel (A), for example, showing a cyclonic pattern centered over the south-western US does not necessarily mean that this is caused by the appearance of a well-defined, anomalous cyclone. While the pattern of the differences is cyclonic, it could be due to the weakening of an anticyclone? The description would have a stronger physical basis if the changes were related to changes in the strength of position of well-recognized climatological features.*

*7. Lines 248 -250: 'In addition, a strong low-pressure system in the middle and upper troposphere over the western US during positive ASO anomaly events (Fig. 6) suggests downwelling flow in the region.' Similar to the concerns about the interpretation of Lines 236 – 240, there is a direct link made between a pattern of changes and the appearance of a particular meteorological feature.*

**Response: We thank the reviewer for this comment. Figure RR10 shows the climatology of April horizontal wind vectors. The circulation over the western United States is controlled mainly by westerlies (no significant anticyclonic circulation). This means that the cyclonic anomaly in Fig. 4 of the revised manuscript is unlikely to be caused by weakening of an anticyclone.**

**We are also aware of the problem associated with this paragraph, and we have rewritten it as follows (please see the lines 199–232 in the revised manuscript):**

**"*Figure 3 shows the correlation coefficients between March ASO anomalies and April zonal wind variations at 200, 500, and 850 hPa, respectively. The spatial distribution of significant correlation coefficients over the North Pacific exhibits a tripolar mode with a zonal distribution at 200 and 500 hPa; i.e. a positive correlation in the high and low latitudes in the North Pacific and a negative correlation in mid-***

*latitudes. This implies that the increase in ASO can result in enhanced westerlies in the high and low latitudes of the North Pacific but weakened westerlies in the mid-latitudes, corresponding to the weakened Aleutian Low in April, and vice versa for the decrease in ASO. The Aleutian Low acts as a bridge connecting variations in ASO and circulation anomalies over the North Pacific (Xie et al., 2017a). At 850 hPa, the anomalous circulation signal in the low latitudes of the North Pacific has weakened and disappeared. It is evident that the anomalous changes in the zonal wind over the North Pacific can extend westward to East Asia. Xie et al. (2018) identified the effect of spring ASO changes on spring precipitation in China. Note that the weakened westerlies in the mid-latitudes and the enhanced westerlies at low latitudes can also extend eastward to the western United States. This kind of circulation anomaly corresponds to two barotropic structures; i.e., an anomalous anticyclone in the Northeast Pacific and a cyclone in the southwestern United States at 500 hPa and 200 hPa. Coincidentally, the northwestern United States is located to the north of the intersection of the anticyclone and cyclone, corresponding to convergence of the airflow at high levels, which may lead to downwelling in the northwestern United States, and vice versa for negative March ASO anomalies.*

*To further validate our inference regarding the response of the circulation in the western United States to ASO changes, we analyze the differences between April horizontal wind anomalies during positive and negative March ASO anomaly events at 200, 500, and 850 hPa (Fig. 4). As in the increased ASO case, the difference shows an anomalous anticyclone in the Northeast Pacific and an anomalous cyclone in the southwestern United States. This kind of circulation anomaly over the southwestern United States enhances cold and dry airflow from the North American continent to*

*the North Pacific, reducing the water vapor concentration in the air over the western United States and possibly reducing April precipitation in the northwestern United States. In addition, the northwestern United States is located to the north of the intersection of the anticyclone and cyclone, suggesting downwelling flow in the region."*

[Figure]

**Figure RR10. Climatology (1984–2016) of April horizontal wind vectors from NCEP2 at 200 hPa (a), 500 hPa (b) and 850 hPa (c).**

*8. Lines 251 – 262: While I can understand how changes in vertical velocity (w) are coherent with the large-scale changes in circulation, the text in this paragraph makes a direct link between changes in w from the NCEP2 reanalysis and changes in convective precipitation. For example, at lines 253 – 255: 'When the March ASO increases, tropospheric convective activity in the northwestern US (115°–130° W)*

*weakens, corresponding to anomalous downwelling.' Can a direct link between convective precipitation and changes in monthly-average vertical velocity be made? I think the authors would need to support this statement with citations to previous work. I am also some-what sceptical about the general direction of the argument, which appears to be trying to link the circulation changes to precipitation changes. Is convective precipitation an important fraction of precipitation in the north-west US in March-April? I would have thought the precipitation changes shown in Figure 1 are a much more straight-forward reflection of changes in orographic precipitation related to the decrease in wind and (presumably) moisture transport?*

**Response: We thank the reviewer for this comment. There was indeed a problem with the description in this paragraph; in particular, the use of the phrase "convective activity" is inaccurate. As can be seen from the Responses to #6 and #7, the large-scale circulation caused by ASO anomalies may lead to upwelling or downwelling in the northwestern United States. Upwelling (downwelling) favors (inhibits) precipitation. This view is often expressed in papers analyzing precipitation (e.g., Kang et al., 2011). The relevant literature has been cited in the revised manuscript. In addition, Figure RR11 shows a significant negative correlation ($r = -0.72$) between vertical velocity (Pa/s) and precipitation anomalies in the northwestern United States in April. This also demonstrates that upwelling (downwelling) in the northwestern United States favors (inhibits) precipitation. This paragraph has been rewritten as follows in the revised manuscript (lines 233-244):**

*"Figure 5a shows a longitude–latitude cross-section of differences in April vertical velocity anomalies averaged over 1000–500 hPa between positive and negative March ASO anomaly events. When the March ASO increases, anomalous downwelling is found in the northwestern United States (115°–130° W). This situation may inhibit precipitation in the northwestern United States in April. Figure 5b depicts the longitude–height cross-section of differences in April vertical velocity averaged over 43°–50°N between positive and negative March ASO anomaly events, which further shows that anomalous downwelling over the United States when the*

*ASO increases. Based on the above analysis, the circulation anomalies in the northwestern United States associated with positive March ASO anomalies may inhibit the formation of local precipitation in April, and vice versa for that with negative March ASO anomalies."*

[Figure]

**Figure RR11. Standardized time series of April precipitation and vertical velocity (Pa/s) (averaged over 1000–500 hPa) from 1984 to 2016. Both quantities are averaged over the area 43°–50°N, 115°–130°W, and the vertical velocity is multiplied by –1 for ease of comparison. The seasonal cycle and linear trend were removed from the original datasets. Precipitation is from GPCP, vertical velocity from NCEP2.**

**References:**

**Kang, S. M., Polvani, L. M., Fyfe, J. C., and Sigmond, M.: Impact of Polar Ozone Depletion on Subtropical Precipitation, Science, 332, 951–954, doi:10.1126/science.1202131, 2011.**

*9. Lines 267 – 268: The WACCM experiments detailed in Table 1 show that the perturbed ASO simulations vary ozone by +/- 15% between 30N and 90N. How realistic is this perturbation compared with the estimates from SWOOSH and GOZCARDS datasets? Perhaps a figure of the zonal-average difference could be included for the composite positive and negative ASO years? At high latitudes a +/-15% variability does not sound too large, perhaps even a bit small, but a +/- 15% change at 30N seems quite large.*

**Response: We thank the reviewer for this comment. Figure RR12 shows the composite zonal mean ozone anomalies (as a percentage) during positive and negative ASO anomalies events from the SWOOSH and GOZCARDS datasets. As noted by the reviewer, during positive (negative) ASO anomaly events the**

stratospheric ozone anomalies are larger (smaller) than 15% at mid and high latitudes, but smaller (larger) than 15% at lower latitudes. In Fig. RR12, ozone changes are about 15% over most of the region 30°–90°N at 300–30 hPa. To keep the experiment simple, we have increased or decreased ozone throughout the region uniformly in the simulations. In principle, the simulation forced with composite ozone anomalies in Fig. RR12 is the best option. Since the simulated results with uniform changes in ASO are in line with observations, we will not rerun the experiments in this work. However, in future work we will use composite ozone changes as external forcing.

[Figure]

**Figure RR12. Composite ozone anomaly percentage (%) during positive (a, c) and negative (b, d) ASO anomaly events, based on SWOOSH (a, b) and GOZCARDS (c, d) ozone data from 1984 to 2016. See Table 2 in the revised manuscript for the definition of ASO anomaly events.**

*10. Line 275: Beginning here, the results from the WACCM simulations are presented. Figures 9, 11 and 13, which show the differences between the WACCM experiments do not have any indication of the statistical significance. All of the other difference plots did have some manner of denoting statistical significance at the 90% level and these*

*three plots should as well.*

**Response: Thanks for the comment. The statistical significance test is added for the three figures.**

[Figure]

**Figure RR13. Differences between experiments R3 and R2 in terms of April (a) precipitation (mm/day) and (b–d) zonal wind at 200, 500, and 850 hPa, respectively. Dots denote significance at the 95% confidence level.**

[Figure]

**Figure RR14. Same as Fig. RR13, but for the difference between experiments R5 and R4.**

[Figure]

**Figure RR15. Same as Fig. RR13, but for the difference between experiments R7 and R6.**

---

## Author Comment (AC3) · 13 Sep 2018

The comment was uploaded in the form of a supplement:
https://www.atmos-chem-phys-discuss.net/acp-2018-414/acp-2018-414-AC3-supplement.pdf

---

## Author Response (AR2)

**Responses to Referees' Comments**

**Effects of Arctic stratospheric ozone changes on spring precipitation in the northwestern United States (acp-2018-414)**

Xuan Ma, Fei Xie, Jianping Li, Xinlong Zheng, Wenshou Tian,
Ruiqiang Ding, Cheng Sun, and Jiankai Zhang

December 2018

**Response to the Editor**

*1. Based on the detailed reviewer comments, the authors have expanded and very much improved their revised manuscript. I am very grateful to the reviewers for their advice. However, while referee #2 concludes that all previous comments have been sufficiently addressed and recommends publication of the manuscript with only a few minor modifications, referee #1 is not convinced and suggests rejection of the manuscript. The main concern of referee #1 is that the model responses over the northwestern United States due to stratospheric ozone forcing are not statistically significant and the response due to ozone changes alone is of a different sign as the observed correlation. I agree with referee #1 on this point. However, I do feel that this study contains valuable results that justify publication in Atmos. Chem. Phys.*

*In order to be acceptable for publication, I ask the authors to take the additional comments of the referees into account. In particular also to arrive at a more balanced and appropriate discussion of the role of ASO vs. SST variations in forcing of northwestern US precipitation anomalies. The revised discussion should in particular reflect the concerns raised by referee #1. As suggested by referee #2, a more explicit link to the coupled model study of Xie et al. (2017) would help to improve the discussion of ASO vs. SST forcing.*

**Response: We are very grateful to the editor for the positive comments on our work and for giving us another chance to address the problems that the reviewers felt were still present.**

**The figure showing the key model result in this study has been redrawn. Actually, zonal wind anomalies at 200 hPa and 500 hPa are significant at the 95% confidence level over the western United States. In terms of the physical connection, the significant U anomalies at 200 hPa and 500 hPa over the western United States are likely to cause significant changes in precipitation in the western United States.**

**To further confirm the possible influence of ASO on precipitation in the northwestern United States, a transient experiment (1955–2005) based on the atmosphere–ocean coupled WACCM4 model (this links to the study of Xie et al. 2017) has been added in the revised manuscript to answer the question of whether the ASO by itself can cause the Victoria SST mode in the North Pacific**

**and rainfall anomalies in the northwestern United States. The answer is yes.**

**The leading relationship between ASO and precipitation in the northwestern United States is evident in observations, time-slice experiments, and the transient experiment with specified ozone. All these conclusions show that our results are reasonable and credible.**

**More details are given in the Responses to reviewers 1 and 2.**

*2. The statement that dynamical coupling affects primarily the Atlantic sector while ozone changes affect primarily the Pacific sector does not seem to be well supported by other studies. I suggest to de-emphasize this point unless there is specific and robust evidence.*

**Response: Thanks for the comment.**

**In the Response to reviewer 1, we show a recent result from Ivy et al. (2017), who investigated the effect of ASO on Northern Hemisphere surface climate using only observations. A very strong and significant ASO-related signal can be found in the North Pacific sector in their Figure 3a and b. It agrees well with our result that ASO can influence the North Pacific sector. The papers (Smith et al. (2014) and Calvo et al. (2015)) pointed out by reviewer 1 only use model simulations. Our result is based on both observations and simulations.**

**In addition, we show that the ASO-related signal in the North Atlantic sector occurs in March; the ASO-related signal in the North Pacific sector occurs in April. In this study, we focus on the influence of ASO on circulation anomalies in April. This is why our result is different from those of Smith et al. (2014) and Calvo et al. (2015), who focus on the March–April average. If we use the March–April circulation, we also find significant ASO-related anomalies in the North Atlantic.**

**The statement in the original manuscript is indeed not precise. In the revised manuscript, this statement has been amended. However, our results are well supported by other studies. We have given a very detailed answer to this question. Please refer to the Response to Reviewer 1.**

*3. Please take into account also the additional minor comments of both referees. In addition, a reference to the HadISST data set should be included. Why is in one case (Fig. 8) HadISST used and in another case (Fig. 9) CESM SST forcing data?*

**Response: Thanks for the comment. The reference (Rayner et al. 2003) has been cited in the revised manuscript. The CESM SST forcing data are based on HadISST data but they have different resolution. The SST anomalies in Figure 9 are used to force the simulation. Thus, CESM SST forcing data are used.**

**Reference:**

**Rayner, N. A., D. E. Parker, E. B. Horton, C. K. Folland, L. V. Alexander, and D. P. Rowell, 2003: Global analyses of sea surface temperature, sea ice, and night marine air temperature since the late nineteenth century. *J. Geophys. Res.*, 108, 4407.**

**Response to Referee 1**

*I thank the authors for their consideration of the reviewers' comments. They have largely improved many of the cosmetic issues with the original manuscript, but many of the major scientific flaws of the original manuscript remain. Furthermore, the statistical significance contours added to the model results show that the key model results of this study that are foundation for the authors' conclusions in the abstract are not statistically significant. So, unfortunately, I cannot recommend publication of this manuscript for the following reasons:*

**Response: Once again, we thank the reviewer for taking the time to assess the manuscript, highlight important issues, and provide helpful comments and suggestions to improve the manuscript. Although the reviewer suggested rejecting the manuscript, we have given new and detailed responses to the reviewer's new comments to address the problems that the reviewer felt were still present in the study. We hope to get another chance and that the reviewer can judge our work again.**

**Figure 10, the key model result in this study, has been redrawn (shown below as Figure R1) to show the differences between the R5 experiment (forced by ASO+SST anomalies) and R4 experiment in terms of April precipitation (mm/day) and zonal wind at 200, 500, and 850 hPa. To make the significant region clearer, this map focuses on North America. The black (white) dots denote significance at the 90% (95%) confidence level in this figure. Actually, the zonal wind anomalies at 200 hPa and 500 hPa are significant at the 95% confidence level over the western United States (Figure R1b and c) and all the anomalies (including U and precipitation anomalies) are significant at the 90% confidence level in the western United States. In terms of physical connection, the significant U anomalies at 200 hPa and 500 hPa (Figure R1b and c, 95% confidence level) in the western United States are likely to cause significant changes in precipitation in the northwestern United States.**

**In order to further confirm the possible influence of ASO on precipitation in the northwestern United States, a transient experiment R8 (1955–2005) based on the atmosphere–ocean coupled WACCM4 model has been added in the revised manuscript to confirm that the ASO by itself can cause the Victoria SST mode in**

the North Pacific and rainfall anomalies in the northwestern United States. Note that the ozone forcing in the experiment is specified; it is derived from the CMIP5 ensemble mean ozone output. Please refer to R8 in Table 1 in the revised manuscript for a detailed description of the experiment. Figure R2 shows the correlation coefficients between the specified March ASO variations and simulated April 500 hPa U, SST, and precipitation anomalies for the period 1955–2005. The significant and leading effects of the specified ASO anomalies on 500 hPa U, the Victoria mode in the North Pacific, and rainfall anomalies in the northwestern United States are well captured (Fig. R2). As the ozone forcing in the experiment is specified, the relationships between ASO and U and SST and precipitation could only be caused by ASO influencing U, and then U influencing SST and precipitation; the ASO changes are completely independent of the polar vortex.

The leading relationship between ASO and precipitation in the northwestern United States can be found in observations, time-slice experiments (R1–7), and a transient experiment with specified ozone (R8). Table 1 in the revised manuscript gives details of experiments R1–8. All these results strengthen the main conclusion of the article.

Figure R2 and relevant content have been added in the revised manuscript. Please see Lines 336–355 and Figure 12.

[Figure]

Figure R1. Differences between experiments R5 and R4 in terms of April (a) precipitation

**(mm/day) and (b–d) zonal wind at 200, 500, and 850 hPa, respectively. The black (white) dots denote significance at the 90% (95%) confidence level. A detailed description of the experiments may be found in Table 1 in the revised manuscript.**

[Figure]

**Figure R2. Correlation coefficients between the specified March ASO variations and simulated anomalies of April U (a), SST (b), and precipitation (c) for the period 1955–2005 based on the transient experiment R8. Regions above the 95% confidence level are dotted. The seasonal cycle and linear trend were removed from all quantities before correlation.**

*1.  Figure 10, which presents the authors' central conclusion (both Arctic ozone and SST forcings are necessary to reproduce the observed signals), does not actually show statistically significant precipitation and circulation signatures over the Pacific Northwest. In my view, this casts some serious doubt on the robustness of the observed relationship documented in this study, if models cannot faithfully reproduce the observed relationship. The most statistically significant features in the model results are in Fig. 7, which are driven by ozone anomalies only and are of the*

*opposite sign to those in the observations.*

**Response: Thanks for the comment. Please refer to above Response.**

*2. The authors are incorrect in stating that midwinter stratospheric anomalies couple down to the troposphere and impact the North Atlantic sector, but that springtime stratospheric anomalies instead impact the North Pacific sector (lines 67–71). Smith et al. (2014) and Calvo et al. (2015) have shown that Arctic stratospheric ozone anomalies primarily affect the Atlantic sector during spring (see Fig. 4 of Smith et al. 2014; see Figs. 3–4 of Calvo et al. 2015). As I stated in my original review, the authors need to provide a convincing explanation for why their patterns are focused in the North Pacific sector, instead of the North Atlantic sector (as would be anticipated from many previous studies).*

**Response: Thanks for the question. The statement in Lines 67–71 in the original manuscript is not precise. It has been modified in the revised manuscript.**

**Figure 4 of Smith et al. (2014) indeed showed that the ASO significantly influences only the North Atlantic sector. However, they obtained their results from a model that does not contain complete stratospheric processes—the CAM3 model. This issue is also pointed out by Calvo et al. (2015), who repeated the study using the WACCM4 model. In figure 3 of Calvo et al. (2015), the influence of ASO is significant in both the North Atlantic sector and the North Pacific sector (Figure R3 below). Note that the results from Smith et al. (2014) and Calvo et al. (2015) are based entirely on simulations. Recently, Ivy et al. (2017) investigated the effect of ASO on Northern Hemisphere surface climate using observations. A very strong and significant ASO-related signal can be found in the North Pacific sector in Figure 3a and b of Ivy et al. (2017) (shown as Figure R4 below). It agrees well with our result that ASO can influence the North Pacific sector.**

**Figure R5a, c, and e shows the relationship between March ASO and March 500 hPa U anomalies based on composite, correlation and regression analyses, while Figure R5b, d and f shows the corresponding relationship between March ASO and April 500 hPa U anomalies. Strong and significant ASO-related signals can be found in the North Pacific and North Atlantic sectors in March. However, the strong and significant ASO-related signal can only be found in the North Pacific sector in April. Ivy et al. (2017) investigated the effect of ASO on March–**

April average circulation. This is why they obtained significant ASO-related signals in both the North Pacific and North Atlantic. In Figure R6, we show the relationship between March ASO and March–April 500 hPa U anomalies. Significant ASO-related signals are also found in both the North Pacific and North Atlantic.

The above analysis at least suggests that ASO may affect the circulation in both the North Pacific and North Atlantic sectors. However, this study focuses on the impact of March ASO on April precipitation in the northwestern United States; ASO is more likely to influence precipitation in the northwestern United States by influencing circulation over the North Pacific Ocean. Therefore, in our study, we mainly analyze the influence of March ASO on the April circulation over the North Pacific Ocean rather than its effect on the circulation over the North Atlantic Ocean.

In the revised manuscript, this statement has been revised as follows (please see Lines 67–71):

*"Similar to the effects of winter stratospheric dynamical processes on the tropospheric North Atlantic Oscillation and the incidence of extreme weather events (Baldwin and Dunkerton, 2001; Black et al., 2005, 2006, 2009), the depletion of spring ASO can cause circulation anomalies that influence the tropospheric North Atlantic and North Pacific sectors."*

[Figure]

Figure R3. Figure 3 of Calvo et al. (2015). The composite differences of zonal mean zonal wind (m s$^{-1}$) at 500 hPa between the April and May averages of the LOW and HIGH ozone years during 1985–2005. Stippling indicates significant differences at the 95% confidence level.

[Figure]

**Figure R4. Figure 3 of Ivy et al. (2017). Differences (colors) in March–April zonal wind at 500 hPa (m s⁻¹) and sea level pressure (hPa) from MERRA between years with low March ozone and years with high March ozone. The contour lines show the climatological mean winds in the top panel; contours are every 5 m s⁻¹ ( . . . , –10, –5, 5, 10, . . . ). Hatching denotes differences that are statistically significant at the 95% level.**

[Figure]

**Figure R5. (a) Differences in composite March 500 hPa U wind anomalies between positive and negative ASO anomaly events. The ASO anomaly events are selected based on Table 2 in**

the manuscript. **(c) Correlation coefficients between March ASO index and March 500 hPa U wind variations. (e) Regression coefficients of March U wind variations regressed onto the March ASO. ASO is taken from SWOOSH and U winds from NCEP2 for 1984–2016. Before performing the analysis, the seasonal cycle and linear trend were removed from the original dataset. Dots denote significance at the 95% confidence level, according to Student's *t*-test. (b, d and f), Same as (a, c and e), but for March ASO and April 500 hPa U anomalies.**

[Figure]

**Figure R6. Same as Figure R5a, c and e, but between March ASO and March–April 500 hPa U anomalies.**

**The variations of the March polar vortex are the intermediate step linking March ASO anomalies and April precipitation changes in the northwestern United States. Therefore, the correlation between the March polar vortex and April precipitation variations is strong.**

**Here, we divide the variations of the March polar vortex strength (SPV) (Zhang et al. 2018) into two parts: one related to ASO ($SPV_{ASO}$) and one unrelated to ASO ($SPV_{NO\_ASO}$); $SPV=SPV_{ASO}+SPV_{NO\_ASO}$. Figure R7 shows that the correlation coefficients between March $SPV_{ASO}$ and April precipitation variations are significant in the northwestern United States, but are not significant between March $SPV_{NO\_ASO}$ and April precipitation variations. Figure R7 illustrates that, as far as the relationship between March polar vortex and April precipitation variations is concerned, only the March polar vortex variations related to ASO are closely linked to precipitation variations in the**

**northwest United States.**

[Figure]

**Figure R7. (a) Correlation coefficients between the March –SPV$_{ASO}$ (the strength index of SPV is defined by Zhang et al. 2018) and April precipitation variations. (b) As for (a), but between March –SPV$_{NO\_ASO}$ and April precipitation variations. SPV$_{ASO}$ and SPV$_{NO\_ASO}$ are calculated by regressing the March SPV index onto the March ASO index and removing the regressed values from the original SPV. Dots denote significance at the 95% confidence level, according to Student's *t*-test. The long-term linear trend and seasonal cycle in all variables were removed before the correlation analysis. The ASO data are from SWOOSH, and precipitation from GPCP.**

*4. The SST anomalies associated with Pacific Northwest precipitation variations and ASO variations closely resemble those associated with ENSO and the PDO (see Fig. R5 in authors' responses). While the authors have presented some correlation analyses in their responses, I'm not entirely convinced that the observed signals in this manuscript aren't being aliased by concurrent ENSO and PDO variability. It would be nice to show that the observed signals still exist if only neutral ENSO years are sampled, as very large El Niño and La Niña events occur within their composites (Table 2).*

**Response: Thanks for the comment. It is a good idea to check whether the**

precipitation anomalies in the northwest United States are related to ENSO. Figure R8 shows the differences in composite April precipitation anomalies between positive and negative ASO anomaly events during neutral ENSO years. The precipitation anomalies in the northwestern United States agree well with the previous result. Removing the effect of ENSO events completely does not affect the results. Please note that these anomalies are not significant. However, this should be due to the small number of cases.

[Figure]

**Figure R8. Differences in composite April precipitation anomalies between positive and negative ASO anomaly events during ENSO neutral years in 1984–2016. The seasonal cycle and linear trend were removed from the original dataset. The ASO anomaly events are selected based on Table R1.**

**Table R1. Selected ASO anomaly events during neutral ENSO years for 1984–2016.**

| Positive March ASO anomaly events | Negative March ASO anomaly events |
|---|---|
| 2001, 2004, 2010 | 1995, 1996 |

**Response to Referee 2**

*Having reviewed the initial submitted version of the manuscript, I will not provide a summary of the manuscript here. My only significant concern with the revised version is that the authors have not completely shown that the circulation and precipitation anomalies are caused by ASO anomalies alone. The new Figure 12 comparing correlations with SPV and ASO, if anything, strengthens the idea that anomalies in ozone and anomalies in the stratospheric circulation reenforce each other. I am not sure this point really comes through in the tone of the discussion. But at some point it becomes an argument about semantics and I think the reader can decide for themselves.*

*I think the authors have sufficiently addressed the concerns raised by the reviewers on the first version here. I might add that the work presented here is much more easily accepted and put into context when combined with the coupled model simulations presented in Xie et al. (2017). The authors may wish to develop a more direct link with the Xie et al. work and stress that the coupled model simulations presented in Xie et al. (2017) were able to produce SST anomalies that were somewhat like the Victoria mode anomalies from the observed SSTs used here. This point of what causes the SST anomalies, which are imposed in the experiments here, is an integral part of the argument about the role of ASO anomalies and should be strengthened here.*

**Response: Once again, we thank the reviewer for taking the time to assess the manuscript and we sincerely appreciate the reviewer's new comments. We have revised the manuscript carefully according to the reviewer's comments and suggestions.**

**Following the comment, a transient experiment (1955–2005) based on the atmosphere–ocean coupled WACCM4 model, which is shown in Xie et al. (2017), has been added in the revised manuscript to confirm that the ASO by itself can cause the Victoria SST mode in the North Pacific and rainfall anomalies in the northwestern United States. Note that the ozone forcing in the experiment is specified; it is derived from the CMIP5 ensemble mean ozone output. Please refer to R8 in Table 1 in the revised manuscript for a detailed description of the experiment. Figure RR1 shows the correlation coefficients between the specified**

March ASO variations and simulated April 500 hPa U, SST, and precipitation anomalies for the period 1955–2005. The significant and leading effects of the specified ASO anomalies on 500 hPa U, the Victoria mode in the North Pacific, and rainfall anomalies in the northwestern United States are well captured (Fig. RR1). As the ozone forcing in the experiment is specified, the relationships between ASO and U and SST and precipitation could only be caused by ASO influencing U, and then U influencing SST and precipitation; the ASO changes are completely independent of the polar vortex.

The leading relationship between ASO and precipitation in the northwestern United States can be found in observations, time-slice experiments (R1–7), and a transient experiment with specified ozone (R8). Table 1 in the revised manuscript gives details of experiments R1–8. All these results strengthen the main conclusion of the article.

Figure RR1 and relevant content have been added in the revised manuscript. Please see Lines 336–355 and Figure 12.

[Figure]

**Figure RR1. Correlation coefficients between the specified March ASO variations and simulated anomalies of April U (a), SST (b), and precipitation (c) for the period 1955–2005 based on the transient experiment R8. Regions above the 95% confidence level are dotted. The seasonal cycle and linear trend were removed from all quantities before correlation.**

*Minor comments:*

*Line 74-75: I think 'Comparing with the effect of the winter stratospheric dynamical processes...' should perhaps be 'Similar to the effects of winter stratospheric dynamical processes...'*

**Response: Thanks for the suggestion. The sentence has been revised according to this comment. Please see Line 67 in the revised paper.**

*Lines 154-156 are difficult to understand, particularly because the WACCM4-GHG scheme has not been defined.*

**Response: Thanks for the comment. The following sentence has been added in the revised manuscript. See Lines 147–149 in the revised manuscript.**

*"**WACCM4-GHG: The chemistry is specified in this scheme; i.e., the volume mixing ratios of forcings of O3, CO2, CH4, N2O, CFC11, CFC12 and so on are prescribed .**"*

*Lines 353 – 355 'Second, the ASO changes generate SST anomalies over the North Pacific...' should really have a reference to Xie et al. (2017).*

**Response: Thanks. The reference has been added.**

*Lines 402 – 405 'Such circulation anomalies force an anomalous cyclone in the western United States in the middle and upper troposphere, which likely enhances cold and dry airflow from the North American continent to the North Pacific...' At least in the climatological monthly average the winds over the northwestern United States continue to be from the west, bringing moisture in from the Pacific. The anomalous cyclone is not a reversal of the winds, but a reduction in the winds. I do not believe you have shown any westerly transport of dry air from the North American continent to the North Pacific – this would require winds from the east. Not just a reduction in westerly winds. I believe a much more justifiable interpretation of the changes is a reduction in the transport of moisture inland from the Pacific.*

**Response: Thanks very much for the suggestion. This sentence and the relevant content have been modified in the revised suggestion as follows (See Lines 229–234 and Lines 368–373):**

*"**The climatological wind over the northwestern United States blows from west to east, bringing moisture from the Pacific to the western United States. Such circulation anomalies force an anomalous cyclone in the western United States in the middle and upper troposphere, which reduces the climatological wind. It would decrease the water vapor concentration in the air over the northwestern United States.**"*

*Figure 8 – Can the authors check that the revised Figure 8 actually shows*

*significance at the 95% level. To my eye Figure 8 in the revised version looks identical to Figure 10 in the original submission where significance was plotted at the 90% level. The significance is denoted with dots and I am not able to find a single dot that has changed between the original Figure 10 and the revised Figure 8 – even in places where a single, isolated dot appears by itself.*

**Response: Thanks for checking this. In Figure 10 of the original submission, the significance was plotted at the 95% confidence level. Only this figure used a 95% confidence level, but the other figures used a 90% confidence level in the original submission. We are very sorry for the misunderstanding caused by this inconsistency. Please see Figure RR2 below; this was Figure 10 of the original submission.**

[revised manuscript text omitted]